# Analysis of somatic mutations in whole blood from 200,618 individuals identifies pervasive positive selection and novel drivers of clonal hematopoiesis

Nicholas Bernstein[1,4], Michael Spencer Chapman [2,3,4], Kudzai Nyamondo [2,3], Zhenghao Chen[1], Nicholas Williams [2], Emily Mitchell[2,3], Peter J. Campbell [2], Robert L. Cohen [1] ✉ & Jyoti Nangalia [2,3] ✉

Human aging is marked by the emergence of a tapestry of clonal expansions in dividing tissues, particularly evident in blood as clonal hematopoiesis (CH). CH, linked to cancer risk and aging-related phenotypes, often stems from somatic mutations in a set of established genes. However, the majority of clones lack known drivers. Here we infer gene-level positive selection in whole blood exomes from 200,618 individuals in UK Biobank. We identify 17 additional genes, *ZBTB33, ZNF318, ZNF234, SPRED2, SH2B3, SRCAP, SIK3, SRSF1, CHEK2, CCDC115, CCL22, BAX, YLPM1, MYD88, MTA2, MAGEC3* and *IGLL5*, under positive selection at a population level, and validate this selection pattern in 10,837 whole genomes from single-cell-derived hematopoietic colonies. Clones with mutations in these genes grow in frequency and size with age, comparable to classical CH drivers. They correlate with heightened risk of infection, death and hematological malignancy, highlighting the significance of these additional genes in the aging process.

Human cells accumulate somatic mutations, leading to an evolving tapestry of clones throughout our tissues as we age[1–10]. The inferred mechanism in replicating tissues is that a stem cell gains a mutation providing a fitness benefit, leading to a clonal expansion due to selection rather than drift[11–13]. Clones with increased fitness from specific mutations can certainly drive cancer[14–16], but expanded clones can influence other diseases both directly, as in chronic liver disease, and through indirect mechanisms, such as in blood[17–21].

In the past decade, genomic sequencing of blood samples has revealed that clonal hematopoiesis (CH) is common in elderly individuals with apparently normal hematopoiesis, with large-scale retrospective studies identifying associations of CH with hematological

malignancies, cardiovascular disease and all-cause mortality[22]. Initial estimates identified CH in >10% of those over the age of 70 when screening for mutations in a known set of genes[23]. However, the prevalence of CH is highly dependent on the sensitivity of sequencing assays, with very small CH clones reported in most individuals over the age of 50 when using highly sensitive sequencing[24]. Bulk approaches generally detect one to two small clones in individuals. Using single-cell sequencing approaches, dozens of parallel clonal expansions can, in fact, be found in blood in all individuals by the seventh to eighth decade of life, with most expansions lacking known driver mutations[25,26]. Similarly, CH identified on the basis of the presence of passenger mutations within clones that lack known driver mutations has been shown to account for

[1]Calico Life Sciences LLC, South San Francisco, CA, USA. [2]Wellcome Sanger Institute, Hinxton, UK. [3]Wellcome-MRC Cambridge Stem Cell Institute, Jeffrey Cheah Biomedical Centre, University of Cambridge, Cambridge, UK. [4]These authors contributed equally: Nicholas Bernstein, Michael Spencer Chapman. ✉e-mail: rlc@calicolabs.com; jn5@sanger.ac.uk

the majority of CH in blood[10,27,28]. As a result, there are ongoing efforts to comprehensively map the drivers of CH[29] to better understand clonal selection and aging phenotypes of blood.

The UK Biobank (UKBB) provides a large cohort with which to assess gene fitness effects and their associated health effects[30]. In this Article, we exploited the idea that a gene can be identified as under positive selection if one finds an enrichment of nonsynonymous mutations compared to neutral synonymous mutations within that gene's coding sequence[11] and comprehensively examine 200,618 UKBB exomes of blood-derived buffy coat for the presence of positive selection leading to clonal expansions. We validate our findings in 10,837 genomes from single-cell-derived hematopoietic cells and examine the associated clinical phenotypes and outcomes, as reported here.

## Results

### Global positive selection in 200,618 whole blood exomes

We analyzed whole blood exome sequencing from 200,618 UKBB individuals aged 40–70 years (Extended Data Fig. 1a) to identify somatic mutations in blood. Following variant calling using Mutect2 (ref. 31) and stringent filtering to remove artifacts and germline variants (Methods), we initially identified 52,701 putative coding somatic variants across 38,211 individuals (Supplementary Table 1 and Extended Data Fig. 1b). As expected, the fraction of individuals with somatic mutations, their variant allele frequency (VAF) and number of somatic mutations per individual increased with age (Fig. 1a). Using the normalized ratio of nonsynonymous to synonymous somatic mutations (dN/dS, R package dNdScv) across all variants, as well as on a per gene basis, we were able to distinguish genes and specific mutations under purifying, neutral and positive selection[11] (Methods, Fig. 1b and Supplementary Table 2). Globally, we found that, across all types of nonsynonymous coding mutation, the dN/dS ratio was 1.13 (95% confidence interval (CI) 1.11–1.16; Fig. 1b), suggesting that one in every eight (CI 7–10) nonsynonymous mutations detected in this dataset was under selection. Specifically, 1 in every 8–11 missense mutations, 1 in every 4–5 truncating mutations and 1 in ~3 splicing mutations (predominantly affecting *DNMT3A*; Supplementary Table 2) showed evidence of positive selection in blood. Positive dN/dS was found in both young and older individuals (Extended Data Fig. 1c–e), suggesting that the rate of entry of somatic mutations under selection may not be significantly different over this age range. These data validate our recent findings of pervasive positive selection on somatic mutations from >3,000 single-cell-derived whole genomes from a small number of healthy individuals[25] and extend these observations to blood of the UK population, as represented by samples collected by the UKBB.

### Novel fitness-inferred CH drivers are common and increase with age

Of the set of 74 genes typically used to identify CH from recently published large population studies[17,32] (Supplementary Table 3), 14

were under positive selection as measured by dN/dS in UKBB (Supplementary Table 2). This includes the most frequently mutated and recognizable drivers of CH such as mutated *TET2*, *DNMT3A*, *ASXL1*, *PPM1D*, *JAK2*, *TP53*, *SRSF2* and *SF3B1* (Fig. 1c) and also less frequently mutated *BRCC3*, *PHIP*, *CBL*, *KDM6A*, *GNB2* and *GNAS*. Genes within this canonical set for CH (Supplementary Table 3) found to be under positive selection we call 'classical fitness-inferred drivers'. *U2AF1* may be missing from this set due to recognized issues with the hg38 reference assembly genome[33]. We identify a high dN/dS ratio for missense variation in *DNMT3A* and *TET2*, but not for *ASXL1* and *PPM1D* (Supplementary Table 2), in agreement with the known mutation landscape in these drivers[34]. We initially identified 18 genes with large significant dN/dS ratios not in the canonical set of 74 CH genes (Supplementary Table 2). To ensure the additional genes identified were not a reflection of our mutation calling strategy, we validated our approach by independently calling somatic mutations in the additional and classical CH genes using Shearwater[35] and retesting for selection using dN/dS (Extended Data Fig. 1b). A total of 96.1% of variants identified by Mutect2 were independently called by Shearwater (Supplementary Tables 4 and 5), with all additional and classical CH genes also under strong positive selection with the sole exception of *DUSP22*. We called the remaining set of 17 genes 'new fitness-inferred drivers' of CH, to distinguish them from the classical set of CH genes, as shown in Fig. 1c,d (Supplementary Table 6). These genes—*BAX*, *CCL22*, *CCDC115*, *CHEK2*, *IGLL5*, *SH2B3*, *SIK3*, *SPRED2*, *SRCAP*, *SRFS1*, *MAGEC3*, *MTA2*, *MYD88*, *YLPM1*, *ZBTB33*, *ZNF234* and *ZNF318*—included novel genes, recently reported candidate drivers of CH in independent datasets, and some previously reported in association with malignancy (Supplementary Table 6). Mutations in these genes had dN/dS ratios between 5 and 660 (indicating that there were 5–660 times more nonsynonymous mutations than expected by chance for these genes) (Fig. 1c and Supplementary Table 2). Many previously reported genes associated with CH were not found to be under significant positive selection in UKBB (Supplementary Table 2), such as *RUNX1*, *PTEN* and *CUX1*, which may reflect their mutation infrequency, their lower prevalence in healthy individuals compared to those with hematological malignancy, or the sensitivity of sequencing in UKBB. Such genes within the canonical set of CH (as defined in refs. 17,32; Supplementary Table 3) that were not under positive selection in UKBB we call 'classical non-fitness-inferred drivers'.

Overall, 23% of UKBB (47,026 individuals) had a detectable mutation in a classical or new CH gene. A total of 51,264 somatic variants were found in classical fitness-inferred ($n$ = 45,770) and classical non-fitness-inferred ($n$ = 5,494) CH genes, and an additional 5,294 variants were found in new fitness-inferred CH genes. Non-'DTA' (*DNMT3A*, *TET2* and *ASXL1* mutated) CH was boosted by >50%. As expected, both the VAF and population frequency for classical drivers of CH increased with age in the UKBB (Fig. 1e,f). Crucially, new fitness-inferred CH gene mutation frequency also increased with age (Fig. 1e, $\beta$ = 0.00015, $P$ value

**Fig. 1 | Pervasive selection in whole blood exomes in UKBB. a**, Exome-wide somatic mutation frequency, VAF and mutation counts in individuals increase with age. The error bars represent 2× standard error of the mean. The smoothed line represents a second-degree polynomial fit of the actual data, and the shading represents the CI. $N$ = 200,618. **b**, Left: dN/dS is the normalized ratio of nonsynonymous to synonymous mutations. dN represents the rate of nonsynonymous mutations per nonsynonymous site, and dS represents the rate of synonymous mutations per synonymous site. A dN/dS of ~1 is expected under neutrality. Genes with a dN/dS ratio >1, taking into account a trinucleotide-specific mutation rate, indicates the gene is under positive selection ('fitness inferred', FI). HSCs with a mutation under positive selection will clonally expand to result in CH. Right: global positive selection in blood detectable at missense, essential splice site and truncating mutations (comprising nonsense substitutions and frameshift insertions/deletions). Nonsynonymous mutations comprise missense and nonsense single base substitutions. The error bars

represent the 95% CI of the dN/dS parameter for that mutation type. $N$ = 52,701 mutations. **c**, Classical fitness-inferred (FI) CH genes (in blue), Classical non-fitness-inferred (non-FI) genes (in green), and new fitness-inferred (FI) CH genes (in orange) representing both novel genes and several recently reported. The graph shows the dN/dS ratio for nonsense and/or missense variants >1, $q$ value <0.1, plotting the maximum dN/dS value. **d**, New fitness-inferred CH genes (in orange) alongside classical fitness-inferred CH genes (in blue), and the types of nonsynonymous mutation they are under positive selection for. **e,f**, The frequency of individuals (**e**) and mutation log(VAF) (**f**) for new and classical FI CH genes and classical non-FI CH genes versus age. The error bars represent the 2× standard error of the mean. The smoothed line represents a second-degree polynomial fit of the actual data, and the shading represents the CI. $N$ = 200,618 individuals. **g**, The number of individuals in UKBB with detectable somatic mutations in driver genes associated with CH. **h**, The number of individuals carrying CH conferring variants per gene.

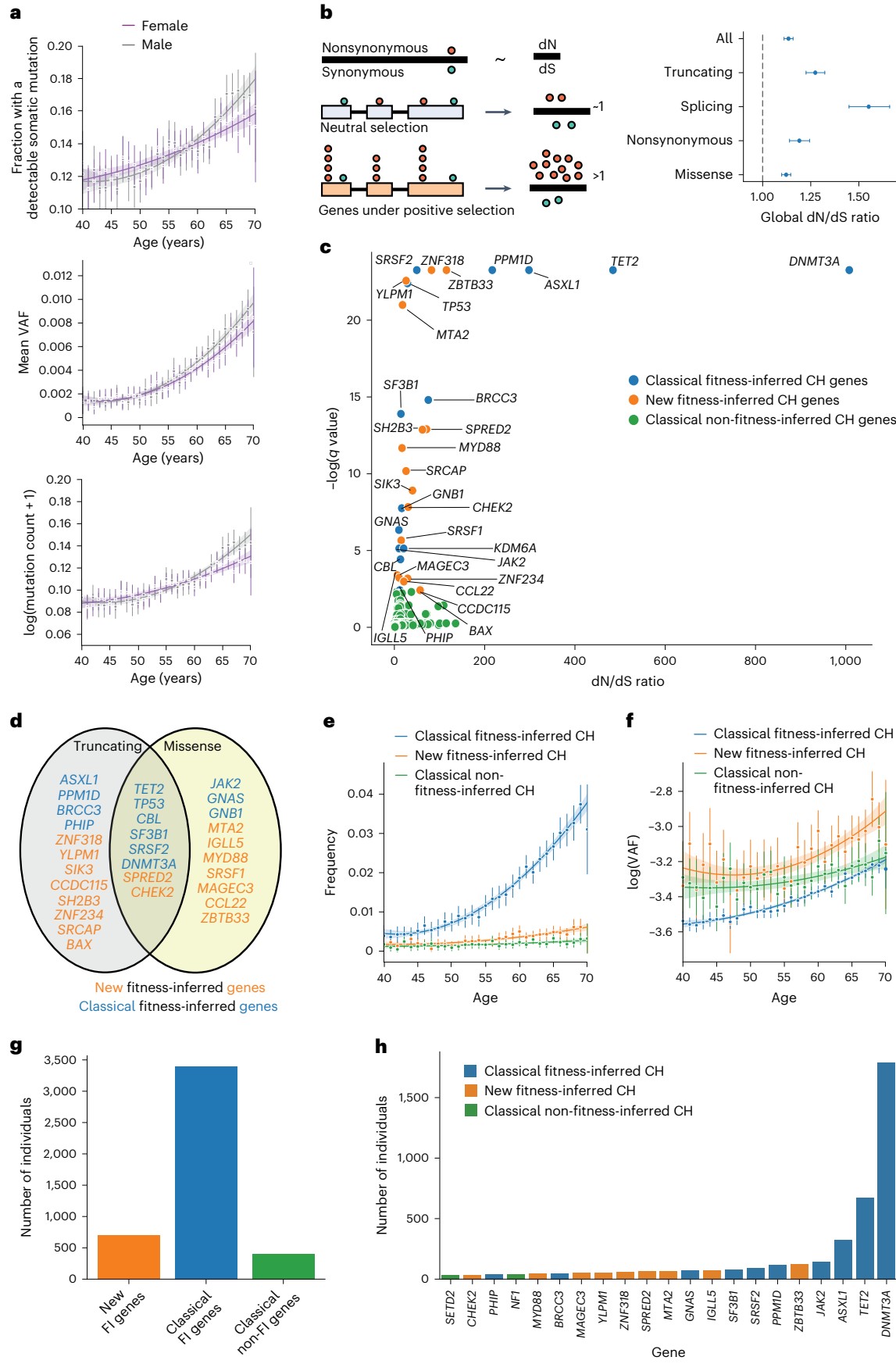

<0.001) and showed a strikingly similar VAF distribution to classical fitness-inferred CH genes ($\beta$ = 0.00852, P value <0.001; Fig. 1f).

When restricting to only larger CH clones (VAF > 0.1), 1.8% of the UKBB exome cohort (n = 3,565) had at least one classical CH mutation, similar to other studies of this age range and sequencing sensitivity[10,32]. Of these, classical fitness-inferred drivers were present in 3,228 individuals and classical non-fitness-inferred drivers affected 371 individuals, with 34 individuals carrying mutations in both CH categories. A total of 681 variants (VAF > 0.1) in new fitness-inferred drivers were identified in 660 individuals (Fig. 1g), the majority of whom (93%, n = 613) did not otherwise harbor mutations in classical CH genes, thus representing an 18% increase to the cohort of individuals with large clone (VAF > 0.1) CH in UKBB.

On a per gene basis, new fitness-inferred drivers were among the most common associated with CH status in individuals in the UKBB (Fig. 1h), with many genes showing distinct mutation landscapes (Fig. 2a). Of these, *BAX*, *CHEK2*, *SH2B3* and *MYD88* mutations were also recently identified as candidate markers of CH in blood from patients with other tumors[29], with *CHEK2* mutations associated with chemotherapy exposure[29,36], and *BAX* mutations reported following BCL2 inhibitor therapy in chronic lymphocytic leukemia (CLL)[37].

Several of the genes identified have been reported in myeloid or lymphoid malignancies. Mutations in *MYD88* are well recognized in Waldenstroms macroglobulinemia and diffuse large B cell lymphoma but also CLL and IgM monoclonal gammopathy of uncertain significance[38,39]. *CCL2* mutations have been reported in chronic lymphoproliferative disorder of natural killer cells[40], and *IGLL5* mutations in CLL[41] and diffuse large B cell lymphoma[42]. It is therefore likely that these mutations derive from the lymphoid compartment in blood. *MTA2* and *MAGEC3* mutations have been reported rarely in acute myeloid leukemia[10,43] and T cell acute lymphoblastic leukemia[44] respectively, where their significance is uncertain, and *SH2B3* mutations have been reported in myeloproliferative neoplasms[45].

Four genes (*ZNF318*, *YLPM1*, *SRCAP* and *ZBTB33*) were also recently identified as new drivers of CH in whole blood exomes from the ExAC database[46], with mutational landscapes similar to those in UKBB. Truncating mutations in *SIK3*, *SPRED2* and *ZNF234*, as well as missense mutations in *SRSF1*, have not been previously described in blood but were under positive selection in UKBB. Most new CH genes are expressed throughout the hematopoietic compartment, with *IGLL5* and *CCL22* gene expression restricted to B cells[47] (Extended Data Fig. 2a).

To appreciate how these additional fitness-inferred genes drive CH in the context of more commonly studied CH mutations, and which regions of these genes may be relevant for conferring fitness (Fig. 2b), we inferred mutation-specific fitness effects as described recently[12] for variants identified >20 times in UKBB (regardless of clone size; Extended Data Fig. 1b). We find that several of these new fitness-inferred driver genes confer some of the strongest fitness effects (Fig. 2b). For example, mutations such as *MTA2* p.Asp289Gly and p.Asp293Gly, *SPRED2* p.Val56Phe and *SRSF1* p.Glu60Asp confer comparable fitness effects to the hotspot variant *DNMT3A*^R882H and other drivers of CH[12], providing a clonal advantage that corresponds to an excess hematopoietic stem cell (HSC) division rate of 15–20% per year (Fig. 2b). Mutations

in the SANT domain of *MTA2*, required for recruitment of HDAC1 and nucleosome remodeling, confer particularly strong fitness (Fig. 2d). Interestingly, the most commonly mutated new fitness-inferred driver gene site, *MYD88* p.Leu273Pro (also commonly referred to as Leu265Pro when protein annotation uses transcript ENST00000396334), was not among sites conferring strong selection (Fig. 2b), and recurrent mutations at this site might be observed because of an intrinsically high mutation rate (Fig. 2c) rather than strong selection.

### Additional CH drivers under selection in blood colonies

To further validate these additional fitness-inferred drivers and identify the cell types affected, we looked for corresponding mutations in several large datasets we have recently published comprising whole-genome sequencing (WGS) of single-cell-derived hematopoietic colonies from individuals with healthy hematopoiesis and blood cancers[13,25,26,48–50]. Clonal WGS data do not suffer from the same artifacts as bulk exome sequencing data, thereby also providing robust orthogonal validation. The combined data comprised WGS from a total of 10,837 individual HSC, myeloid and lymphoid single-cell-derived colonies, from 50 individuals with either healthy hematopoiesis or blood cancer. This includes 10,202 colonies derived from myeloid progenitors and stem cells from healthy aging individuals[25,26,48], cord blood[25], human fetal hematopoiesis[25,49], individuals who have undergone allogeneic stem cell transplantation[50], individuals with myeloproliferative neoplasms[13], additional myeloid malignancies such as therapy-related acute myeloid leukemia, chronic myeloid leukemia and essential thrombocythemia), and 635 B/T cell-derived lymphoid colonies[48]. In total, 150 nonsynonymous variants were identified in 16 of the 17 new fitness-inferred CH drivers across these individuals (Supplementary Tables 7 and 9).

Within the myeloid datasets (both healthy hematopoiesis and hematological disease), we found 127 variants within 15/17 new fitness-inferred driver genes (Supplementary Table 7), with the majority (n = 102) of mutations detected in colonies from healthy individuals (Supplementary Table 10). Ten genes had evidence of positive selection by dN/dS (*CHEK2*, *SRCAP*, *ZNF318*, *ZBTB33*, *MAGEC3*, *SPRED2*, *SIK3*, *YLPM1*, *BAX* and *SH2B3*, q value <0.1, restricted hypothesis testing) with the mutation landscapes resembling those found in the UKBB data (Supplementary Table 8). Mutations that are either rarely acquired, or only selected for in a subset of individuals, may not have been validated by dN/dS due to the far lower number of individuals in the clonal WGS dataset (17 healthy individuals, 10 allogeneic stem cell transplant recipients and 23 individuals with blood cancers).

Within the lymphoid clonal WGS data, among other variants in new fitness-inferred CH genes, there were 15 nonsynonymous mutations in *IGLL5*—a gene found at the immunoglobulin lambda locus—found in three healthy individuals (Supplementary Table 9), and this was the only gene found here to be under selection by *dNdScv*. This suggests that somatic mutations within the lymphoid compartment are driving their occurrence in bulk whole blood exome sequencing in UKBB. Intriguingly, 14/15 mutations in *IGLL5* occurred in memory B cell-derived colonies, even though, overall, memory B cells accounted for <12% of the lymphoid dataset. Overall ~15% of memory B cell colonies

**Fig. 2 | New CH gene-specific mutation landscapes and fitness effects. a,** Location and type of mutation across the gene body for new drivers of CH. Filled black circles, missense mutations; filled white circles, truncating mutations (frameshift indel or nonsense mutations). **b,** Twenty-five recurrently mutated sites with the highest estimated fitness effects include several mutations in new CH driver genes. The error bars for the inferred fitness effect and mutation rate parameters are the CI. The number of observations of VAF for each mutation used to infer parameters is presented in Supplementary Table 4. **c,** Ten recurrently mutated sites with the highest estimated mutation rate. Of note, *MYD88* Leu273Pro, which lies outside the SANT domain, has one of the highest mutation rates within the dataset but did not have a significantly increased fitness effect.

The error bars for the inferred fitness effect and mutation rate parameters are the CI. The number of observations of VAF for each mutation used to infer parameters is presented in Supplementary Table 4. **d,** Fitness estimates for *MTA2* plotted across the gene body, which primarily localize to the SANT domain. The error bars for the inferred fitness effect and mutation rate parameters are the CI. The number of observations of VAF for each mutation used to infer parameters is presented in Supplementary Table 4. Amino acid positions are colored by the gene elements shown in the key. Orange shading refers to new fitness-inferred genes of CH; classical fitness-inferred genes of CH are shown in blue, and classical non-fitness-inferred CH genes are shown in green.

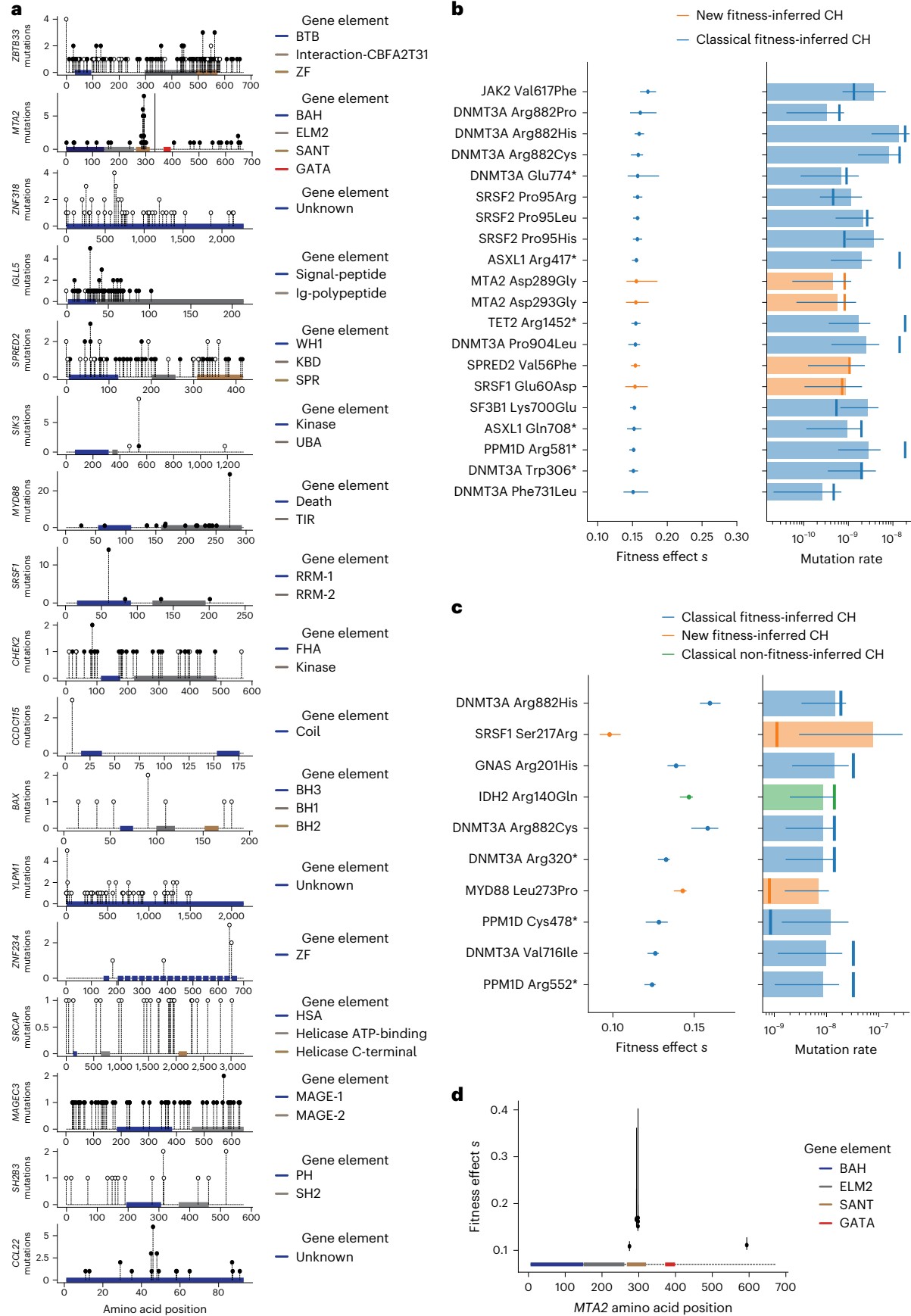

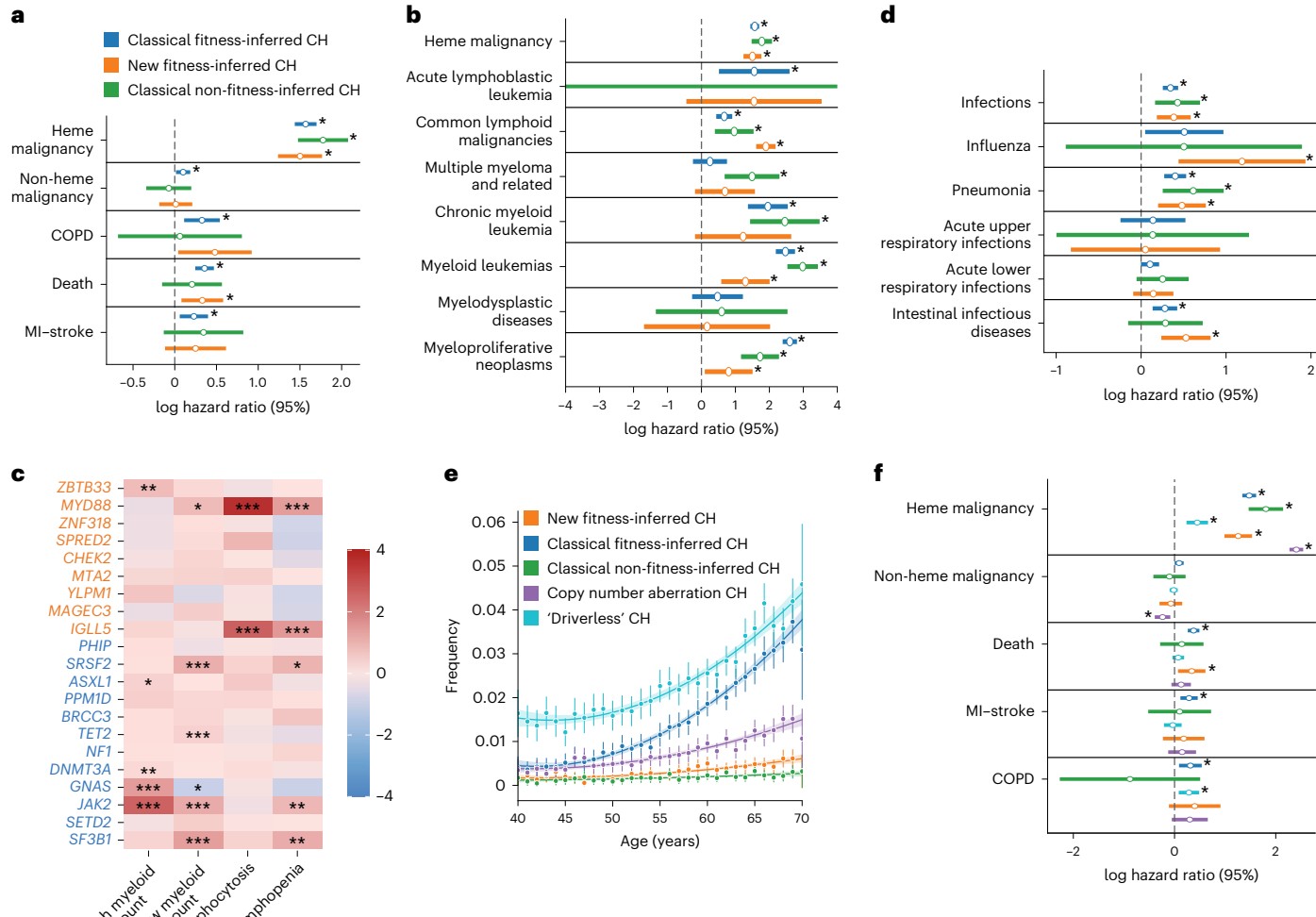

**Fig. 3 | Clinical outcome associations with new drivers of CH. a,b,d,** Cox-proportional HRs for a range of poor health outcomes against CH categories with CIs of log(HR) plotted. *N* = 200,618 individuals. Different categories of poor health outcomes are shown in **a**, hematological malignancies are shown in **b**, and infections are shown in **d**. *P < 0.05. Heme, hematological; COPD, chronic obstructive pulmonary disease; MI, myocardial infarct. **c**, Complete blood count associations of commonly mutated fitness-inferred driver genes. The stars are based on the FDR-adjusted *P* values from logistic regression (**P* value >0.01 to <0.05, ***P* value >0.001 to <0.01, ****P* value <0.001). **e**, The incidence of 'driverless' CH and autosomal mosaic copy number variants with age alongside other CH

categories. The error bars represent the 2× standard error of the mean incidence. The smoothed line represents a second-degree polynomial fit of the actual data, and the shading represents the CI. *N* = 200,618 individuals. **f**, log(HR) with the 95% CI plotted for different categories of CH across poor health outcomes. *N* = 200,618 individuals. The orange shading refers to new fitness-inferred genes of CH, classical fitness-inferred genes of CH are shown in dark blue, classical non-fitness-inferred CH genes are shown in green, CH clones with copy number aberrations on autosomes are shown in purple, and CH clones without detectable candidate driver mutations are shown in light blue.

had *IGLL5* mutations. Both synonymous and nonsynonymous mutations clustered around the N-terminus in B cell-derived colonies from healthy individuals and in whole blood exome sequencing in UKBB (Fig. 2a), a finding also echoed in *IGLL5* mutations in lymphoid neoplasms found in the COSMIC database[34]. This implies that the gene may be subject to highly localized mutational processes—for example, during B cell somatic hypermutation[41]—that could invalidate assumptions underlying the dN/dS methodology. This raises the possibility that mutations in *IGLL5* may be a passenger event during memory B cell expansion rather than driving clonal expansion themselves. No mutations in other suspected lymphoid genes, for example, *MYD88*, were found in this dataset, though it is notable that not all lymphoid compartments were represented within the single-cell-derived lymphoid colonies.

Overall, across >10,000 single-cell-derived clonal hematopoietic cell colonies derived from 50 individuals, we identified 150 somatic mutations in 16 of the 17 new fitness-inferred drivers, with a strikingly

similar pattern of nonsynonymous mutations to bulk whole-exome sequencing data from UKBB. The majority of variants (*n* = 125 of 150) were from myeloid and lymphoid colonies from individuals with healthy hematopoiesis (Supplementary Table 10), with no history of malignancy or chemotherapy exposure, demonstrating that positive selection acts on mutations in new fitness-inferred CH genes during healthy hematopoiesis. Importantly, 10 of the 17 new fitness-inferred CH genes were also under positive selection, as estimated by dN/dS in single-cell-derived colonies from a small set of individuals, validating UKBB findings. Two of the remaining seven genes (*SRSF1* and *MTA2*) did show independent evidence of positive selection based on the distribution of the VAFs of recurrently mutated sites (Fig. 2b). Excluding *IGLL5*, the significance of which remains to be determined, and *MYD88*, a previously reported driver of lymphoid malignancies, there are three remaining genes (*CCDC115*, *ZNF234* and *CCL22*) identified as under positive selection in UKBB that we could not independently validate. While two *ZNF234* variants were found in hematopoietic

colonies, including one nonsense mutation, and a missense mutation in *CCL22* was present in a naive B cell colony, these numbers were too low to display gene-wide positive selection. Incidentally, these three genes were also the least frequently mutated in UKBB, and they should be considered provisional CH genes pending future studies of larger datasets where their potentially increased variant numbers may allow further elucidation.

## Increased risk of hematological neoplasm, death and infection

The UKBB provides continuously updated electronic health records that we used to assess for the impact of clonal expansions associated with new fitness-inferred CH genes on subsequent health events (Supplementary Table 11). Many of the new fitness-inferred CH drivers have been proposed as candidate drivers of CH previously, or are known to be mutated in specific hematological malignancies (Supplementary Table 6), but their clinical associations have not been characterized at a population scale. We considered individuals with CH driver mutations at an allele frequency of >0.1 (Supplementary Table 5) as the clinical effect of CH is known to be attenuated in smaller clones[17,51,52]. Individuals were removed from association analyses if they had chronic obstructive pulmonary disease (COPD), myocardial infarction or stroke, nonhematological malignancy and hematological malignancy events before blood draw for whole-exome sequencing, as well as if there was a mismatch between X chromosome zygosity and reported sex[53]. We performed a Cox-proportional hazard regression to assess for the association of drivers of CH (both new and classical, VAF >0.1) with hematological malignancies. We found that new fitness-inferred driver genes confer a significant hazard for hematological malignancy (Fig. 3a, log(hazard ratio (HR)) 1.5, *P* value <0.001) similar to recognized drivers of CH. This hazard is strongest for common lymphoid malignancies (Fig. 3b, CLM, log(HR) 1.9, *P* value <0.001), driven primarily by mutations in *MYD88* (log(odds ratio (OR)) 2.5, *P* value <0.001) associating with chronic lymphoid malignancy, and *IGLL5* (log(OR) 2.8, *P* value <0.001) associating with acute lymphoblastic leukemia and chronic lymphoid malignancy, with weaker associations with other malignancies (Extended Data Fig. 2b,c). Both genes have been reported as drivers of lymphoid malignancies[38,39,41,42]. The presence of mutations in many new fitness-inferred CH genes were mirrored by changes in peripheral blood counts, with *ZBTB33* mutations associated with increased myeloid counts and *MYD88/IGLL5* mutations associated with abnormal lymphocyte counts (Fig. 3c).

CH has been associated with increased risk for myocardial infarction, ischemic events, COPD and all-cause mortality[17,18,54,55]. Similar to classical fitness-inferred drivers, individuals with new fitness-inferred driver mutations also have a significantly increased hazard for death (Fig. 3a, *P* = 0.01, log(HR) 0.36). Crucially, incidence of poor health outcomes was dependent on clone size, which is a phenomenon also observed in classical CH[55] (Extended Data Fig. 3). Recently, mosaic copy number variants in blood were associated with an increased risk for a wide range of infections[20]. CH driven by nonsynonymous/short indel mutations have not yet been shown to confer an increased risk for these outcomes. We show here that all categorizations of CH associate with increased hazards for infections, intestinal infectious diseases and pneumonia (Fig. 3d).

## The majority of clonal expansions remain unexplained
'Driverless' CH is the occurrence of clonal expansions in blood without a known driver[10,25] and is estimated to drive the majority of clonal expansions in the elderly. We identified individuals with a high mutational burden without any known CH driver (as identified by this study; Methods) or copy number variants (as described in ref. 56). We confirm that 'driverless' CH in UKBB is very common and increases with age as previously seen (Fig. 3e), conferring a significantly increased risk for COPD and hematological malignancies but not all-cause mortality (Fig. 3f).

An ongoing debate is whether 'driverless' CH is caused by somatic driver mutations that are yet to be identified, nongenetic factors (for example, epigenetic changes) or neutral drift. Our dN/dS analysis of mutations >0.11 VAF—indicating that 1/8 of nonsynonymous mutations are under positive selection—suggests that unrecognized driver mutations still remain a large contributor. With our expanded definition of CH, we estimate that we now identify ~50% (CI 43–59%) of the total number of drivers of CH in blood (Supplementary Note 1). This is an increase on previous estimates of positive selection in blood[27] and suggests that we are now 'halfway there' to identifying driver mutations that are able to produce large clones in blood. Since *dNdScv* appears to linearly recover increased numbers of drivers of CH within this cohort (Supplementary Note 2), there would be merit in a future substantially larger study combining genome sequencing of several different population cohorts to identify further genes under positive selection in blood.

## Discussion
Somatic mutation acquisition in human cells occurs stochastically, reflecting tissue-specific rates and mutational processes, with only a minority of these mutations confering a fitness advantage that drives clonal expansion. However, recent studies have suggested that clonal selection on somatic mutations in blood extends beyond that attributable to known driver mutations in a set of <100 genes[10,25,27]. In whole blood exomes from ~200,000 individuals from UKBB, over 10% of nonsynonymous mutations that we identified may be under positive selection. Of note, the high dN/dS ratios in UKBB whole blood exomes compared to other studies[25] could be due to ascertainment bias. First, bulk exome sequencing data will only capture mutations that occurred before the expansion of the most recent common ancestor of a detectable clone, and not any subsequent mutations. Secondly, in the absence of single-molecule sequencing, driver mutations that occurred earlier in life are more likely to be captured than later driver mutation acquisitions due to the greater duration for clonal expansion. Both of these factors could have the impact of inflating the number of driver mutations compared to background mutations. Nevertheless, at a gene-specific level, we identify 17 further candidates in this clonal apparatus. While some of these genes have been recently reported as candidate drivers[46], or identified in the context of concurrent malignancies[29] and therapy[36,37,56], here we show that positive selection on these genes is detectable in unselected populations, within individuals with healthy myeloid and/or lymphoid hematopoiesis, and that those who carry a nonsynonymous mutation in these genes (at VAF >0.1) having significantly increased hazards for a wide range of adverse health outcomes.

In this study, we used one of the many methods available to identify regions within exome sequences that are under positive selection to identify additional driver mutations in whole blood exomes (restricted to those associated with large clones >0.1) together with validation of positive selection in single-cell-derived hematopoietic myeloid and lymphoid colonies. Inclusion of mutations in these fitness-inferred CH genes increases the prevalence of CH (restricted to large clones >0.1) by 18% in the UKBB cohort. Other methods that infer selection coefficients from variant VAFs[12] or mosaic copy number variants[20,56] may lead to more CH driving loci associated with poor health outcomes. Studies of different hematopoietic populations, such as HSCs or specific lymphoid populations not represented in whole blood, may also identify additional genetic sites under positive selection. Nevertheless, it is clear that many drivers of clonal expansion in blood are still unknown, in stark contrast to other tissues such as the esophagus, where our compendium of fitness-inferring somatic mutations appears much more complete[27]. Future efforts enabling characterization of smaller clonal expansions, using both more sensitive and less error-prone sequencing technologies, as well as combined population datasets, will enable a more comprehensive examination of the evolutionary substrate under positive selection in blood.

The pervasive impact of age-associated oligoclonality and gene-specific clonal expansions on health outcomes is increasingly clear. Future research is needed to determine the extent to which aging phenotypes are encoded by the landscape of clonal expansions, in both blood and other self-renewing tissues, to determine the degree to which modulating the burden of somatically mutated clones can improve health outcomes.

## Online content

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

## Methods

### Ethics statement

Informed consent was obtained for all individuals providing samples for hematopoietic colonies under NHS Research Ethics Committee approval 18/EE/0199 and 07/MRE/44. Participants for in-house whole genomes were recruited under Cambridge Blood and Stem Cell Biobank ethics, 18/EE/0199 and 07/MRE/44, following written informed consent, and for published genomes as reported previously[13,25,26,48–50]. UKBB analysis was conducted using the UKBB resource under application 18448.

### Somatic variant calling in UKBB and dNdScv analysis

Putative somatic variants were initially identified using Mutect2 (ref. 31) (broadinstitute/gatk:4.1.3.0) from exome sequencing CRAM files (UKBB resources 23143 and 23144). We used 1000 Genomes, ExAC and gnomAD to remove sequencing artifacts and common germline single-nucleotide polymorphisms (SNPs)[57–59]. Variants were called within target capture regions (UKBB resource 3801) and 100 bp either side and annotated using SNPeff[60] (v4_3) and dbSNP build GRCh38.86. A total of 83,396,753 putative variants underwent quality control filtering to remove (1) variants with a median base quality difference of >5 between the alternate and reference allele, (2) mutation sites showing extreme strand bias ($P < 0.01$, chi-square test) due to overrepresentation of sequencing artifacts[61], (3) variants with features commonly associated with false positives, such as alleles only supported by the end of the read, or reads with excessive edit distance, using FINGs v1.7.1 (ref. 62) (using thresholds as follows: minbasequality 30, zeroproportion 0.05, minmapquality 50, minmapqualitydifference 5, enddistance 10, enddistancemad 3, editdistance 4, maxoaftumor 0.04, maxsecondtumor 0.05, foxog 0.9, snvcluster50 2, snvcluster100 4, repeats 12), (4) indels with 10 or fewer reads supporting the alternate allele[17], and (5) variants in the gene *MUC*, olfactory genes and genes with recurrent synonymous mutations due to high rates of false positivity[63,64]. Variants with VAF <0.11 and <3 reads supporting the alternate allele were also excluded to reduce false positive mutations. This threshold was chosen because our desired number of false positive mutations was <1 for the entire exome cohort. The total number of sites tested for our project is equal to the size of the targeted exome region in UKBB (38,997,831 base pairs[65]) times the number of individuals (200,618). Given a sequencing error in 1/1,000 base pairs, we wanted to find the VAF threshold for the median locus depth, 45 reads, which would lead to <1 false positive somatic mutation from sequencing error. Using the formula Number of false positives = Number of bases tested $x$ sequencing error$^{(\text{median loci depth} \times \text{VAF})}$, led to the choice of a VAF threshold of 0.11. Additional filters were applied to remove (1) variants which appeared germline (binomial test, $P = 0.5$, $n$ represents depth of locus, $k$ represents read count of alternate allele, $P < 1 \times 10^{-4}$), (2) variants with a $-\log_{10}$ population minor allele frequency in gnomAD of <3.35 as this threshold allowed removal of germline variants while retaining known drivers of CH common in gnomAD, such as $JAK2^{\text{V617F}}$, (3) variants that occurred more often than $DNMT3A^{\text{R882H}}$ as well as mutations with more than 50% of their occurrences having a VAF >0.15 (ref. 46), (4) variants in homopolymers regions of length 4 (ref. 66), (5) high depth (>150×) variants, and (6) variants within 50 bp of one another. A total of 52,701 variants remained following the above filtering (Extended Data Fig. 1b). *U2AF1* mutations may have been missing from this list of variants due to recognized issues with mutation calling relating to the hg38 reference assembly genome[33]. We used dNdScv[11] (https://github.com/im3sanger/dndscv) to perform neutrality tests by calculating dN/dS ratios on the 52,701 filtered Mutect2 variants on both a global and a per gene level for different classes of mutations (missense, nonsense, indel and essential splice site). A dN/dS ratio >1 for any type of variation ($q$ value <0.1) was considered evidence of positive selection. To validate both variants and genes under selection, and to gain sensitivity for detecting low VAF clones in putative genes of interest, we next ran the

somatic variant caller Shearwater[35] (v3_11) on all 200,618 UKBB exomes for all classical (Supplementary Table 3) and new fitness-inferred CH genes (Supplementary Table 6). We included uniquely mapped reads with mapping quality >30. CosmicCodingMut.vcf from COSMIC v94 (ref. 34) was used to create a prior of Shearwater. We considered variants to validate our Mutect2 call with a Bayes factor <0.5. Finally, we reran dNdScv only on the intersection of variants (96.1%) that were called by both Mutect2 and Shearwater. All new and classical fitness-inferred driver genes except *DUSP22* were still under significant positive selection using these high-quality validated variant calls. *DUSP22* was removed from subsequent analysis.

### Determining CH status in UKBB and clinical correlates

We began with the union of unfiltered variant calls from Shearwater and Mutect2 to identify which variants would confer CH status (Extended Data Fig. 1b). Shearwater variants were included only if they were recurrently found ($n > 5$) in the Mutect2 call set. We then removed variants with (1) VAF >0.5 unless they were on chromosome X in males, (2) excess strand bias as described above, (3) a $-\log_{10}$ population minor allele frequency <3.35 according to gnomAD, and (4) if >50% of a variant's occurrences had a VAF >0.15. We required variants to have an allele frequency of >0.1 as the clinical effect of CH is known to be decreased in smaller clones[17,51,52]. This provided the final variant list for conferring CH status. We divided CH identifying genes into three categories: (1) classical fitness-inferred drivers, that is, previously described genes in CH[17,32] under positive selection in UKBB; (2) classical non-fitness-inferred drivers, that is, mutations in genes historically associated with CH but not found to be under positive selection in the UKBB; (3) new fitness-inferred drivers, that is, mutations in both novel genes and several recently reported genes that have not routinely been used to identify CH status and under strong positive selection in UKBB. Genes were under differential selection for missense versus nonsense mutation. For example, *MTA2* was under positive selection only for missense variation according to dNdScv, so only missense variants in *MTA2* were considered to confer new fitness-inferred CH status. For nonsense variation, any truncating variant was used to confer new fitness-inferred CH status. We used linear regression to identify relationships between age and frequency of each category of CH as well as log(VAF). The log (VAF) distribution was assumed to be normal, but this was not formally tested. We identified poor health outcome events and their corresponding dates from UKBB data (Supplementary Table 11). Individuals were removed from association analyses if they had COPD, Myocardial infarct-stroke, nonhematological malignancy and hematological malignancy events before blood draw, and if there was a mismatch between X chromosome zygosity and reported sex[53]. We used Cox-proportional hazard regression with new fitness-inferred CH status, classical fitness-inferred CH status, classical non-fitness-inferred status, body fat percentage, pack-years of smoking, age at recruitment, age at recruitment², self-reported sex, hypertension, systolic, diastolic, low-density lipoprotein, high-density lipoprotein, cholesterol and type 2 diabetes, as covariates (Supplementary Table 12). Per gene effects for genes mutated in >30 individuals in UKBB were inferred using logistic regression using Python statsmodels (0.12.2) for each health outcome as described above (Supplementary Table 13).

### Inferring fitness coefficients for recurrently mutated sites

We performed maximum likelihood estimates for the fitness parameter of the 150 most commonly mutated sites in the UKBB, following the method described by Watson et al.[12]. We included all variants at these sites called by Shearwater or Mutect2 with VAF <0.5. Fitness effects were estimated as selection coefficients ($s$), which represents the increased birth rate of an HSC relative to wild-type HSCs as a result of the driver mutation. We estimated $N\tau$, where $N$ represents the total number of HSCs and $\tau$ represents the time in years between symmetric divisions,

and $\sigma$, the standard deviation of the age of the cohort using the most common mutation in our cohort, $DNMT3A^{R882H}$. Our estimated parameters for $N\tau$ (140,000) and $s$ (0.159) were similar to previous estimates[2,12]. We removed variants from consideration where the CI for $s$ was >0.2.

### Determining mosaic CNV and 'driverless' CH status and clinical correlates
Individuals with mosaic CNV were identified from Loh et al.[67]. To identify large 'driverless' CH, we identified individuals with an excessive number of somatic mutations[10] (≥95th percentile, $n$ = 3 somatic mutations). We removed individuals with known driver mutations (VAF >0.02) or autosomal mosaic CNV events. This identified 5,873 individuals with 'driverless' CH. To identify associated clinical outcomes, we used the same set of outcomes and covariates as described above for our Cox-proportional hazard regression for individuals with >0.1 cell fraction of mosaic CNV and all identified 'driverless' CH.

### Gene expression and gene set enrichment
We visualized normalized expression log(transcripts per million sequencing reads) data of sequenced bulk RNA from sorted normal, preleukemic and leukemic cells of the hematopoietic lineage from Corces et al.[47] (GSE74912) plotting mean log(transcripts per million sequencing reads) per cell type for each gene on a heatmap. Gene set enrichment analysis was performed using the $q$-value ranked list of genes from dNdScv to identify significantly enriched (false discovery rate (FDR) cutoff of 0.1) MSigDB hallmark gene sets[68,69].

### dN/dS analysis of mutations from hematopoietic colonies
WGS of single-cell-derived colonies was derived from published[13,25,26,48–50] and unpublished datasets. Mutations from published single-cell colony datasets[13,25,26,48–50] and unpublished datasets were divided by whether they contained predominantly hematopoietic stem cell/multipotent progenitor (HSC/MPP) and myeloid progenitor colonies, or lymphoid colonies. The myeloid and HSC/MPP datasets included those where colonies were grown on methylcellulose agar or were flow-sorted HSC/MPPs. Given that colonies from the same individual may share somatic mutations that occurred in a common ancestor, we only considered unique mutations from each individual. Combined mutation sets across patients were analyzed using the dndscv function from the R package 'dndscv' (https://github.com/im3sanger/dndscv) using default settings except for the following arguments: max_muts_per_gene_per_sample = Inf, use_indel_sites=T, max_coding_muts_per_sample = Inf. This part of the analysis was not aimed at new driver discovery but rather validation of the new genes discovered from UKBB data. Therefore, analysis was restricted to the 17 new genes: $SPRED2$, $MTA2$, $YLPM1$, $ZBTB33$, $ZNF318$, $ZNF234$, $SRSF1$, $IGLL5$, $MYD88$, $SIK3$, $CHEK2$, $MAGEC3$, $CCDC115$, $BAX$, $SRCAP$, $SH2B3$ and $CCL22$, which were set within the 'gene_list' argument of the dndscv function. Genes were considered to have evidence of positive selection if the reported dN/dS $q$ value was <0.1 for missense mutations, truncating variants, all substitutions or indels.

### Peripheral blood count associations with CH
To quantify the impact of new genes associated with CH on peripheral blood counts, we used the complete blood count parameters obtained by the UKBB. These parameters allowed us to categorize individuals into five categories: high myeloid cell parameters, low myeloid cell parameters, lymphocytosis, lymphopenia and normal blood count parameters, as previously described[70]. We then used logistic regression using the Python package stansmodels (0.12.2) to identify the relationship between each peripheral blood count abnormality and the new genes associated with CH using the individuals with normal blood count parameters as the control group. Pack-years smoking, age at recruitment and sex were used as covariates.

### Statistics and reproducibility statement
No statistical methods were used to predetermine sample sizes. No data were excluded from the analysis. The experiments were not randomized and the investigators were not blinded to allocation during experiments and outcome assessment. Data collection and analysis were not performed blind to the conditions of the experiments.

### Reporting summary
Further information on research design is available in the Nature Portfolio Reporting Summary linked to this article.

## Data availability
Individual level data will be uploaded to UKBB in keeping with UKBB's data sharing agreement. Nonindividualized mutation data have been provided in the supplementary tables. Hematopoietic colony-level data are available as follows: (1) healthy hematopoietic colonies (WGS accession numbers EGAD00001007684 and EGAD00001007851), (2) myeloproliferative neoplasm colonies (WGS accession number EGAD00001007714), (3) fetal hematopoiesis colonies (WGS accession number EGAD00001006162), (4) lymphoid colonies (WGS accession number EGAD00001008107), (5) chronic myeloid leukemia colonies (WGS accession number EGAD00001015353), (6) allogeneic stem cell transplant (WGS accession number EGAD00001010872) and (7) single individual with therapy-related acute myeloid leukemia (WGS accession number EGAD00001015339). Source code for figures has been provided as Supplementary Information. Source data are provided with this paper.

## Code availability
Code for analyses is available online at https://doi.org/10.5281/zenodo.10891332 ref. 71 and can also be found at https://github.com/nangalialab/UKBB_ClonalHaem_Novel_Drivers as well as at https://github.com/mspencerchapman/Pervasive_positive_selection_in_blood.

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

## Acknowledgements

We thank N. Rubinstein, E. Sorokin, M. Cule, C. Connelly, E. Melamud and F. Harding for their feedback and guidance. UKBB analysis was funded by Calico Life Sciences LLC and conducted under UKBB application 18448. J.N. is a Cancer Research UK (CRUK) Advanced Clinician Scientist Fellow. K.N. is supported by CRUK and the Wellcome Trust. Work in the Nangalia lab is supported by CRUK, Wellcome core funding at Wellcome Sanger Institute, Alborada Trust and Rosetrees Trust.

## Author contributions

N.B. performed UKBB analysis with Z.C. under supervision of R.L.C. and J.N. R.L.C. conceived the study. R.L.C. and J.N. directed the research. M.S.C. performed analysis of hematopoietic colonies with E.M. and P.J.C. N.B., K.N., M.S.C., R.L.C. and J.N. prepared the manuscript. All authors reviewed the manuscript. R.L.C. and J.N. jointly supervised this work.

## Competing interests

N.B. is an ex-employee of Calico Life Sciences LLC; R.L.C. and Z.C. are employees of Calico Life Sciences LLC. P.J.C. is a cofounder and shareholder of FL86 Inc. The other authors declare no competing interests.

## Additional information

**Extended data** is available for this paper at https://doi.org/10.1038/s41588-024-01755-1.

**Correspondence and requests for materials** should be addressed to Robert L. Cohen or Jyoti Nangalia.

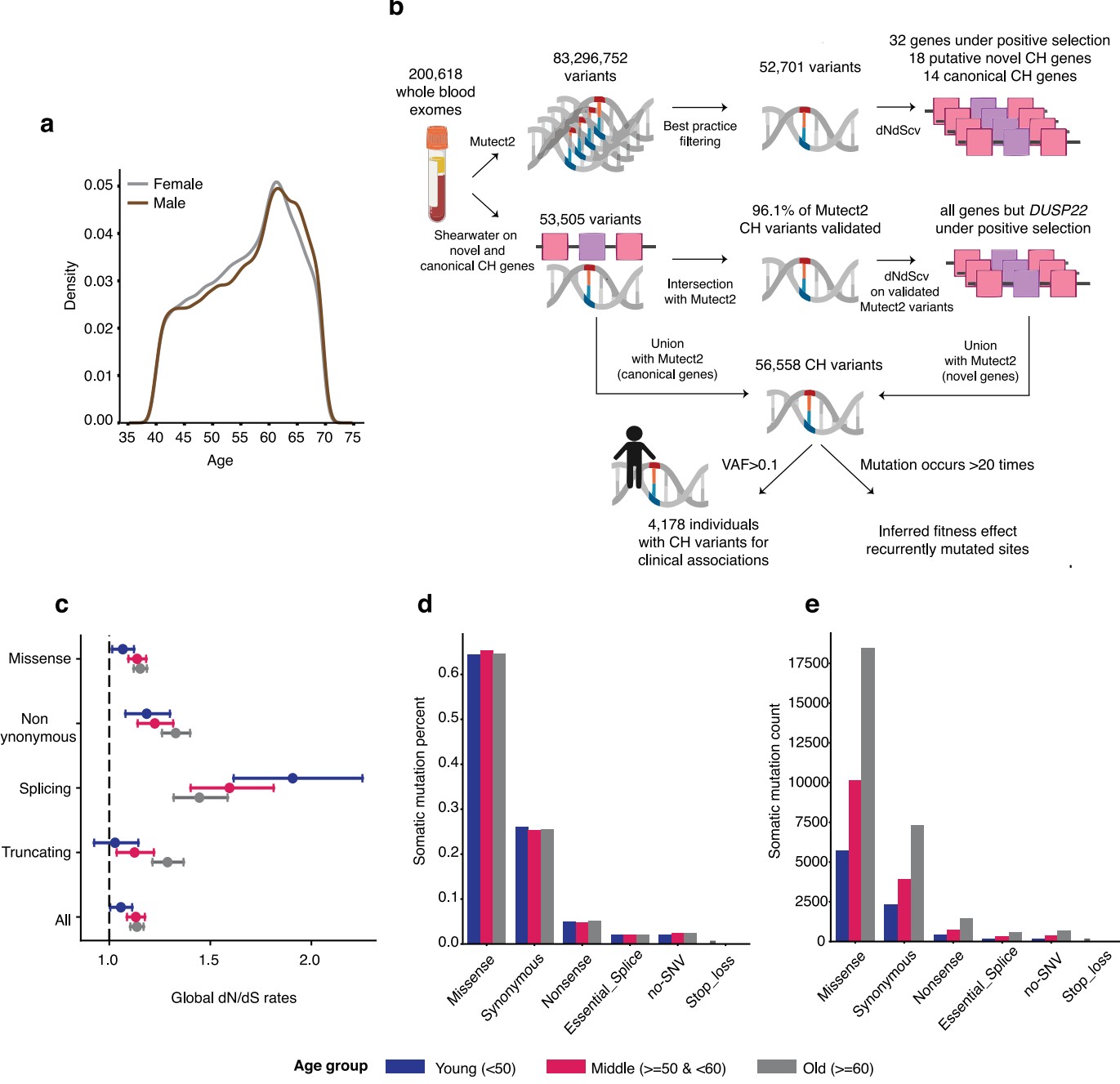

**Extended Data Fig. 1 | Variant calling, global selection and somatic mutations in UK Biobank. a.** Age distribution from UKBB. **b.** Pipeline for identifying exome wide selection on somatic mutations and new genes under positive selection in UKBB. **c.** Global dN/dS estimates per age group. Error bars represent the 95% CI of the dN/dS parameter for that mutation type and age group. N = 52,701 mutations. **d.** Percent of mutation type per age group. **e.** Number of mutations for each mutation type per age group.

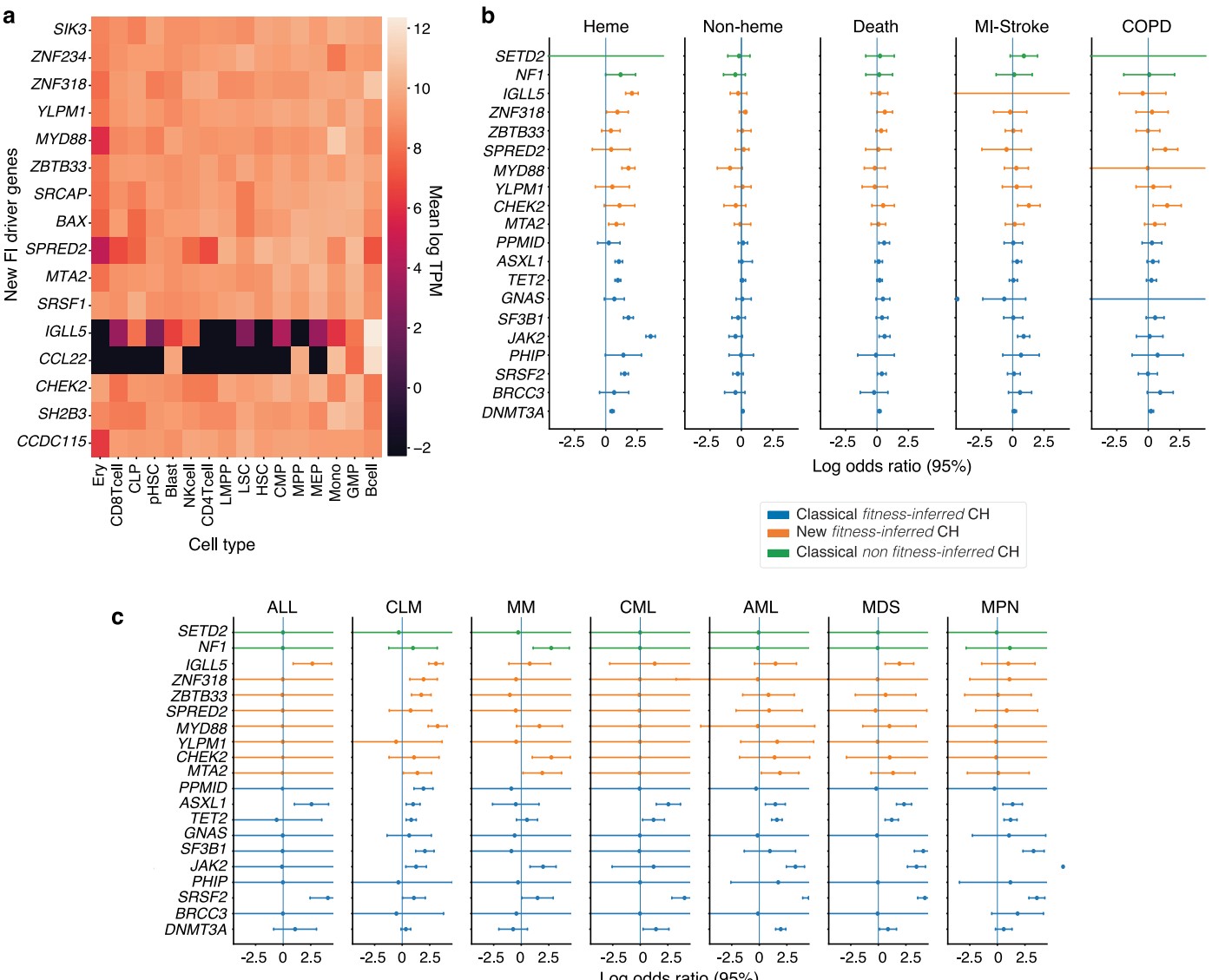

**Extended Data Fig. 2 | Gene expression and poor health outcome associations of recurrently mutated clonal haematopoiesis genes.** Gene expression of new 'fitness-inferred' (FI)-driver genes in haematopoietic compartment. Expression data was taken from Corces et al.[47]. Common lymphoid progenitor, CLP; megakaryocyte-erythroid progenitor cell, MEP; common myeloid progenitor, CMP; granulocyte-macrophage progenitor cell, GMP; lymphoid-primed multipotent progenitor cell, LMPP; preleukemic HSC, pHSC; leukemia stem cells, LSC; erythroid cell, Ery; leukemic blast cell, blasts; multipotent progenitor cell, MPP; Mono, monocyte; NK cell, natural killer cell; HSC, haematopoietic stem cell. **b**, **c**. Logistic regression log odds ratios for a wide range of poor health outcomes for commonly mutated (genes with >35 mutations) driver genes of CH. Error bars represent the 95% CI. N = 200,618 individuals. Note the log odds ratio for *JAK2* mutations and MPN is shown as '>' as it is >5 (5.96, 95% CI 5.56–6.36). Acute lymphoblastic leukaemia, ALL; Common lymphoid malignancies, CLM; multiple myeloma and related, MM; chronic myeloid leukaemia, CML; acute myeloid leukaemia, AML; myelodysplastic diseases, MDS; myeloproliferative neoplasms, MPN; acute upper respiratory infection, AURI; acute lower respiratory infection, ALRI; intestinal infections, II. Orange shading refers to new FI genes of CH; classical FI genes of CH are shown in blue, and classical non FI CH genes are shown in green.

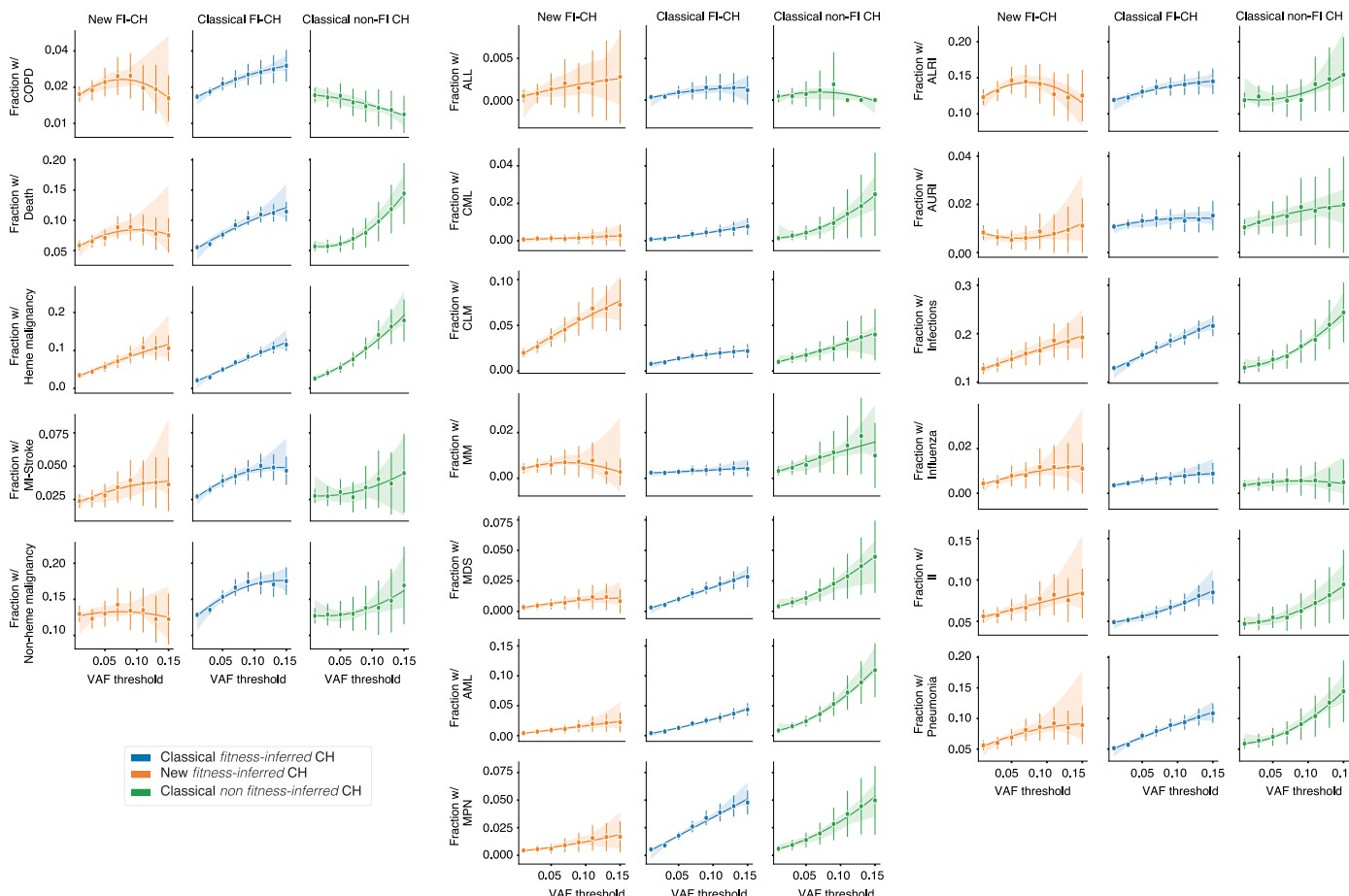

**Extended Data Fig. 3 | Clone size effect on poor health outcomes.** Clone size effect on poor-health outcomes (malignancy, stroke, death, different haematological neoplasms and infections) incidence by CH driver mutation category. Poor health outcomes incidence increases as clone variant allele frequency increases to 0.1 after which the relationship becomes unstable which has been noted previously for *DNMT3A* and *TET2* mutant clones and

cardiovascular risk. Error bars represent the 2*standard error of the mean incidence. The smoothed line represents a second degree polynomial fit of the actual data and the shading represents the 95% CI. N = 200,618 individuals. Orange shading refers to new FI genes of CH; classical FI genes of CH are shown in blue, and classical non FI CH genes are shown in green.

# Reporting Summary

## Statistics

For all statistical analyses, confirm that the following items are present in the figure legend, table legend, main text, or Methods section.

| n/a | Confirmed | |
|---|---|---|
| ☐ | ☒ | The exact sample size (*n*) for each experimental group/condition, given as a discrete number and unit of measurement |
| ☐ | ☒ | A statement on whether measurements were taken from distinct samples or whether the same sample was measured repeatedly |
| ☐ | ☒ | The statistical test(s) used AND whether they are one- or two-sided *Only common tests should be described solely by name; describe more complex techniques in the Methods section.* |
| ☐ | ☒ | A description of all covariates tested |
| ☐ | ☒ | A description of any assumptions or corrections, such as tests of normality and adjustment for multiple comparisons |
| ☐ | ☒ | A full description of the statistical parameters including central tendency (e.g. means) or other basic estimates (e.g. regression coefficient) AND variation (e.g. standard deviation) or associated estimates of uncertainty (e.g. confidence intervals) |
| ☐ | ☒ | For null hypothesis testing, the test statistic (e.g. *F*, *t*, *r*) with confidence intervals, effect sizes, degrees of freedom and *P* value noted *Give P values as exact values whenever suitable.* |
| ☒ | ☐ | For Bayesian analysis, information on the choice of priors and Markov chain Monte Carlo settings |
| ☒ | ☐ | For hierarchical and complex designs, identification of the appropriate level for tests and full reporting of outcomes |
| ☒ | ☐ | Estimates of effect sizes (e.g. Cohen's *d*, Pearson's *r*), indicating how they were calculated |

*Our web collection on statistics for biologists contains articles on many of the points above.*

## Software and code

Policy information about availability of computer code

| Data collection | UKBiobank exome sequencing CRAM files were obtained from UKBiobank resources 23143 and 23144. Variant calling files for single cell derived haematopoietic colonies was available internally at the Sanger Institute, and for published works, also available publically as detailed in the following papers (Mitchell et al, Nature 2022, Williams et al, Nature 2022, Spencer Chapman et al, Nature 2021, Fabre et al, Nature 2022, Machado et al, Nature 2022, Spencer Chapman et al, Blood 2022. Diagnoses of individuals with haematopoietic colony sequencing were as provided by previous publications or as collected following informed consent under NHS Research Ethics Committee approval 18/EE/0199 and 07/MRE/44. |
|---|---|
| Data analysis | Mutect2 (broadinstitute/gatk:4.1.3.0) and Shearwater (v3_11, Gerstung et al, Bioinformatics 2014) were used for variant identification. 1000 genomes, ExAC and gnomAD were used to remove sequencing artefacts and common germline SNPs. Variants were called within target capture regions (UKBB resource 3801) and 100bps either side and annotated using SNPeff (v4_3) and dbSNP build GRCh38.86. Variants with features commonly associated with false positives, such as alleles only supported by the end of the read, or reads with excessive edit distance, were excluded using FINGs v1.7.1. The R package dNdScv (https://github.com/im3sanger/dndscv) was used to detect gene and global level positive selection using default settings except for the following arguments: max_muts_per_gene_per_sample = Inf, use_indel_sites=T, max_coding_muts_per_sample = Inf. COSMIC v94 (Tate et al Nucleic Acids 2019) was used to create a prior for Shearwater (v3_11). Gene set enrichment analysis and gene expression was analysed using data from Corces et al (GSE74912). |

For manuscripts utilizing custom algorithms or software that are central to the research but not yet described in published literature, software must be made available to editors and reviewers. We strongly encourage code deposition in a community repository (e.g. GitHub). See the Nature Portfolio guidelines for submitting code & software for further information.

## Data

Policy information about [availability of data](availability of data)

All manuscripts must include a [data availability statement](data availability statement). This statement should provide the following information, where applicable:

- Accession codes, unique identifiers, or web links for publicly available datasets
- A description of any restrictions on data availability
- For clinical datasets or third party data, please ensure that the statement adheres to our [policy](policy)

> Individual level data will be uploaded to UKBB in keeping with UKBB's data sharing agreement. Non individualised mutation data has been provided in the Supplementary Tables. Code for analyses has been made available online at  https://github.com/nangalialab/UKBB_ClonalHaem_Novel_Drivers and https://github.com/mspencerchapman/Pervasive_positive_selection_in_blood.

## Research involving human participants, their data, or biological material

Policy information about studies with [human participants or human data](human participants or human data). See also policy information about [sex, gender (identity/presentation), and sexual orientation](sex, gender (identity/presentation), and sexual orientation) and [race, ethnicity and racism](race, ethnicity and racism).

| | |
|---|---|
| Reporting on sex and gender | Sex and gender were considered in the study design - where reported gender did not match X chromosome zygosity in UKBB, samples were excluded from further analysis. |
| Reporting on race, ethnicity, or other socially relevant groupings | No hypotheses related to clonal haematopoiesis and race, ethnicity or socially relevant grouping were asked in this study. |
| Population characteristics | UKBiobank population characteristics were as previously described in Bycroft et al Nature 2018. We also analysed 10837 colonies from 50 individuals from pre-birth to >80 years of age: 10,202 colonies derived from myeloid progenitors and stem cells from healthy ageing individuals (Mitchell et al 2022, Fabre et al 2022), , cord blood (Mitchell et al Nature 2022), human foetal haematopoiesis (Spencer Chapman et al Nature 2021),  individuals who have undergone allogeneic stem cell transplantation (Spencer Chapman et al, Blood 2022), individuals with myeloproliferative neoplasms (Williams et al, Nature 2022), additional datasets from additional myeloid malignancies such as therapy-related acute myeloid leukemia, chronic myeloid leukaemia and essential thrombocythaemia), and 635 B-/T-cell derived lymphoid colonies (Machado et al, Nature 2022) |
| Recruitment | For the UKBiobank analysis, recruitment was undertaken by UK Biobank. For the analyses of single-cell derived haematopoietic colonies from individuals with healthy haematopoiesis and haematological malignancies, we used previously published data. Individuals with sequencing data from unpublished datasets were recruited under Cambridge Blood and Stem Cell Biobank ethics, 18/EE/0199 and 07/MRE/44. |
| Ethics oversight | For single cell derived colonies, Ethics oversight was by the Eastern Multi-region Ethics Committee and under Cambridge Blood and Stem Cell Biobank ethics, 18/EE/0199 and 07/MRE/44. |

Note that full information on the approval of the study protocol must also be provided in the manuscript.

# Field-specific reporting

Please select the one below that is the best fit for your research. If you are not sure, read the appropriate sections before making your selection.

☒ Life sciences ☐ Behavioural & social sciences ☐ Ecological, evolutionary & environmental sciences

For a reference copy of the document with all sections, see [nature.com/documents/nr-reporting-summary-flat.pdf](nature.com/documents/nr-reporting-summary-flat.pdf)

# Life sciences study design

All studies must disclose on these points even when the disclosure is negative.

| | |
|---|---|
| Sample size | The full UKBiobank dataset of 200K exomes was used for the study. For single cell derived colonies, we maximised the number of individuals and colonies based on published datasets and additional sequencing available. No sample size calculations were performed, however, we intended on using at least 10,000 single cell derived colonies (or interrogation of effectively 10,000 haematopoietic stem cells) to be sufficiently powered to detect , the dN/dS ratios observed for the additional genes in UKBB, were these mutations also to be found in the colony datasets. |
| Data exclusions | For UKBiobank, individuals were removed from association analyses if they had COPD, MI-Stroke, non-haematological malignancy, and haematological malignancy events prior to blood draw, and if there was a mismatch between X chromosome zygosity and reported gender. For haematopoietic colonies, all samples were considered for dn/ds analysis. |
| Replication | To validate mutations identified by Mutect (broadinstitute/gatk:4.1.3.0) , Shearwater (v3_11) was used for variant calling. dN/dS (dNdSc (https://github.com/im3sanger/dndscv)) was performed on both the Mutect calls, and on the intersection of Mutect and Shearwater validated calls too. Of 18 genes originally identified as under selection using Mutect for variant identification, 17 genes validated when using Shearwater. DUSP22 did not validate and was excluded from further analysis. Further validation was sought by calling identifying mutations in |

novel genes associated with clonal haematopoiesis in whole genomes of single cell derived colonies as detailed in the methods section of the manuscript.

| | |
|---|---|
| Randomization | The analysis or study design was not randomised as no therapy was being tested as part of a clinical trial. |
| Blinding | Data analysis and collection was not performed blind and the investigators were not blinded to the allocation during analysis or outcome assessment. |

# Reporting for specific materials, systems and methods

We require information from authors about some types of materials, experimental systems and methods used in many studies. Here, indicate whether each material, system or method listed is relevant to your study. If you are not sure if a list item applies to your research, read the appropriate section before selecting a response.

## Materials & experimental systems

| n/a | Involved in the study |
|---|---|
| ☒ | Antibodies |
| ☒ | Eukaryotic cell lines |
| ☒ | Palaeontology and archaeology |
| ☒ | Animals and other organisms |
| ☐ | ☒ Clinical data |
| ☒ | Dual use research of concern |
| ☒ | Plants |

## Methods

| n/a | Involved in the study |
|---|---|
| ☒ | ChIP-seq |
| ☒ | Flow cytometry |
| ☒ | MRI-based neuroimaging |

## Clinical data

Policy information about clinical studies

All manuscripts should comply with the ICMJE guidelines for publication of clinical research and a completed CONSORT checklist must be included with all submissions.

| | |
|---|---|
| Clinical trial registration | N/A |
| Study protocol | The study was conducted using UK Biobank under application number 18448. Haematopoietic colony samples were collected under research ethics committee approval 18/EE/0199 and 07/MRE/44. |
| Data collection | Data collection was conducted by UKBiobank. For single cell derived haematopoietic colonies, no clinical data was required from participants beyond the haematological diagnosis which was gathered from the previous publications of the datasets or under Cambridge Blood and Stem Cell Biobank ethics, 18/EE/0199 nad 07/MRE/44. |
| Outcomes | We identified poor-health outcomes events and their corresponding dates from UKBB data as detailed in Supplementary table 10. Individuals were removed from association analyses if they had COPD, MI-Stroke, non-haematological malignancy, and haematological malignancy events prior to blood draw, and if there was a mismatch between X chromosome zygosity and reported gender. No outcomes were measured from haematopoietic colony data. |

