## [Peer Review File · Nature Genetics]

Peer Review Information

Manuscript Title: Analysis of somatic mutations in whole blood from 200,618 individuals identifies pervasive positive selection and novel drivers of clonal hematopoiesis

Corresponding author name(s): Dr Robert Cohen, Dr Jyoti Nangalia

Reviewer Comments & Decisions:

Decision Letter, initial version:

13th Sep 2023

Dear Dr Nangalia,

Your Article, "Pervasive positive selection in blood in 200,618 individuals and novel drivers of clonal haematopoiesis" has now been seen by 3 referees. You will see from their comments below that while they find your work of interest, some important points are raised. We are interested in the possibility of publishing your study in Nature Genetics, but would like to consider your response to these concerns in the form of a revised manuscript before we make a final decision on publication.

To guide the scope of the revisions, the editors discuss the referee reports in detail within the team, including with the chief editor, with a view to identifying key priorities that should be addressed in revision and sometimes overruling referee requests that are deemed beyond the scope of the current study. In this case, we encourage you to address Reviewers' comments in full. Please do not hesitate to get in touch if you would like to discuss these issues further.

We therefore invite you to revise your manuscript taking into account all reviewer and editor comments. Please highlight all changes in the manuscript text file. At this stage we will need you to upload a copy of the manuscript in MS Word .docx or similar editable format.

*2) If you have not done so already please begin to revise your manuscript so that it conforms to our Article format instructions, available here.

*3) Include a revised version of any required Reporting Summary:

Please be aware of our guidelines on digital image standards.

[redacted]

We hope to receive your revised manuscript within four to eight weeks. If you cannot send it within this time, please let us know.

Best wishes,
Chiara

Chiara Anania, PhD
Associate Editor
Nature Genetics
<https://orcid.org/0000-0003-1549-4157>

Referee expertise:

Referee #1: clinical, hematology oncology

Referee #2: WGS, blood, clonal hematopoiesis

Referee #3: clonal hematopoiesis genetics (signed report)

Reviewers' Comments:

Reviewer #1:

Remarks to the Author:

The manuscript from Bernstein and Chapman et al. "Pervasive positive selection in blood in 200,618 individuals and novel drivers of clonal hematopoiesis" reports an analysis of the UK Biobank (UKBB) dataset identifying genes that are clonally expanded in the blood of study participants. They present evidence that these genes are under positive clonal selection. They show that CH with mutations in these genes correlates with poor outcomes with similar to what has been established for CH.

The text is well written overall and the figures are nicely presented. While this study certainly has something to contribute to the field, and the findings are of significant interest, the central advance of this paper is the identification of additional CH genes which is more incremental than innovative at this point in time. Additionally, several technical issues should be addressed:

Major points:

(1) Most of the manuscript is built on concepts of dN/dS and selection coefficient (s) borrowed from evolutionary theory. This is certainly a valuable perspective, however this methodology makes a number of assumptions about the data that may not optimally apply to the data, and as a field we need to be careful about using this measurement/calculcation as an arbiter of truth. The authors appropriately acknowledge the situation of somatic hypermutation in the lymphoid compartment which is one example. In addition that some other issues warrant at least discussion as limitations.

(1a) Peripheral blood VAF does not measure VAF in the cells which acquire and maintain mutant clones (such as HSCs) of the blood population. The dynamics and interpretation are therefore fundamentally different than ie a population of poorly differentiated carcinoma cells in which most have high (?unlimited) replicative potential. One consequence of this is that mutations that improve HSC fitness and replication while also blocking differentiation would be interpreted to be deleterious to fitness (fewer mature daughter cells in the peripheral blood) while mutations that increases propagation only in later stages of differentiation and do not actually alter HSC fitness might be interpreted as "drivers". I applaud the author's use of the single cell derived hematopoietic colony dataset as an orthogonal method of validation which addresses the issue for about half the FI genes, but there are still the others that were not validated and conceptual factors could be at play in addition to the sample size issue discussed.

(1b) Another way in which human blood systems are different from populations with ideal genetic mixing is that total HSC number (not just diversity) and bone marrow cellularity decrease as humans age. Is it appropriate to apply dN/dS to a population with a decreasing number of individuals and

declining habitat?

(1c) How strong is the correlation between dN/dS and # of individuals with a mutation? It looks like a Pearson r would be large based on the dN/dS of genes in figure 1C and number of individuals with mutations in each gene in figure 1H. Which genes could be identified only by this method that we would be blind to if only using number of mutations (obviously we don't need it to see that DNMT3A, TET2, and other highly prevalent genes are under positive selection).

(1d) Could there be a systematic issue relating to how dNdScv interacts with variant calling making it so that exonic regions with either poorer read coverage or that tend to be covered near the ends of reads where filters are more stringent are less likely to be called and could bias interpretation i.e. if they are interpreted as places where the probability of mutation is high but none/few mutations are found?

(1e) I'm surprised that there such an apparently strong MYD88 hotspot mutation is not regarded as strongly positively selected for in the model. The mutation identified is different than L265P which we do clinical testing for in the workup of Waldenstrom's, but is still in a functionally annotated domain that is highly evolutionarily conserved (at least through Danio Rerio). Are there really so many non-synonymous mutations in this region that this hotspot does not look selected for? If so, would a non-coding regulatory element within the DNA sequence whose disruption causes synonymous mutations to have a fitness effect be one possible explanation?

(2) There appear to be very few total mutations in some of the 17 FI-CH genes identified by the authors based on figure 2A. For example CCDC115 appears to be mutated only twice in the entire dataset (or mutated ≈ 2.5 times based on the y-axis labeling). If I am interpreting this correctly, I do not think that 2 mutations is sufficient to calculate meaningful estimates of the synonymous and non-synonymous mutation rates, or of their ratio. Also related to CCDC115 it is curious that both of the mutations are so early in the gene because truncating mutations in this position can often be skipped. Does dNdScv treat all truncations the same regardless of position in the gene? This might be a situation in which that is not a good approximation.

(3) Several of the "17 new genes" are not new. As appropriately discussed by the authors some commonly mutated genes like CHEK2, SRCAP, and MYD88 are already supported by other literature. This is appropriately cited but it would be helpful if the authors can expand on how their study advances understanding of these genes and more specifically focus on the novel CH genes they identified.

(4) It wasn't clear to me that an appropriate correction for multiple hypothesis testing was performed in the analysis and interpretation of hazard ratios in figure 3A,B,D,F.

Minor points:

(1) Abstract text lines 14 and 15: The statement that CH is typically driven by somatic mutations in a small set of genes but that most clonal expansions have unknown drivers seems self-contradicting, or at least isn't easy to understand.

(2) Figures 1A, 1E, and 3E, and extended data Figure 3 show model fits of data. This is very pretty but please show the actual data somehow like for example median and std deviation in 2 year bins. The

model used to fit the data and meaning of error bars (nearly invisible in my printed copy in the main figure panels) should be described in the figure legend.

(3) Main text line 96: "infrequent mutation infrequency" is redundant.

(4) At the bottom of Figure 2D the words "Lorem Ipsum" are printed apparently in error.

(5) The way in which the supplementary tables are enumerated was garbled at least in the combined PDF version I received.

(6) In the interest of transparency and to protect the authors from the appearance of impropriety then if any authors have stock and/or ownership stake in Calico and/or parent company Alphabet that should be disclosed in addition to the employment relationship. If Calico and/or any authors have filed provisional patents to protect intellectual property related to this manuscript that should be disclosed as well.

Reviewer #2:

Remarks to the Author:

In this manuscript, the authors use data from >200k participants in the UKBiobank with whole-exome sequencing (WES) data to infer gene-level selection in the context of clonal hematopoiesis. They identify 17 new genes that are the targets of positive selection. Single-cell WGS data were used to verify that these 17 genes are under positive selection and to identify the stem cells/progenitor cells in which these mutations are present. In so doing, they define a set of "non-canonical fitness-inferred" CH driver mutations that have fitness effects comparable to canonical CH drivers and that also increase in frequency and size with age. This study represents a clever use of the UKBB WES data and dN/dS ratios to expand the set of known CH driver mutations. Given this expanded set of CH drivers, they estimate that "driverless" CH represent about half of all CH cases. As far as I could tell, the study contained no real conceptual or technical flaws. The paper was also very well-written. Therefore, the following are a list of suggestions to strengthen what is already an excellent paper.

In the Introduction it is worth mentioning that the prevalence of clonal hematopoiesis (CH) is highly dependent on what mutant cell fractions (CF) are identifiable. For instance, defining CH as clones with CF > 5% leads to a much different estimated prevalence than if CH is defined as a clone with CF > 0.1%.

Can the authors comment on how the disease status of the 50 individuals with sc-HSC sequencing data may have affected their results?

An additional analysis that may be of interest would be to compare clonal growth rates (estimated via PACER, PMID: 37046083) between the group of individuals with canonical CH drivers and the group of individuals with non-canonical-FI CH drivers.

Can the authors comment on the lack of a statistically significant association between the novel FI-CH and MI-Stroke (in comparison to the Canonical FI-CH mutations)? Is this just a matter of statistical power (i.e., are there many fewer carriers of novel FI-CH mutations than carriers of canonical FI-CH mutations)?

Reviewer #3:

Remarks to the Author:

In "Pervasive positive selection in blood in 200,618 individuals and novel drivers of clonal haematopoiesis", Bernstein and colleagues infer gene-level selection in blood using exome data from the 200k exome release of the UK Biobank. They identify a set of genes under positive selection with similar selection characteristics to the set of widely established CH driver genes. Overall, the analysis is very robust and thoughtfully executed. Several minor aspects of the presentation could use additional attention.

1. It is surprising that based on the fitness effect estimated by the dN/dS method here, DNMT3A R882 is among the most fit mutation, however in prior work co-authored both by this group (eg Fabre, Nature, 2022, Ref 24) and observed in multiple other studies (eg Robertson, Nature Med 2022, Weinstock, Nature 2023), DNMT3A R882 is among the least fit. How do the authors reconcile these observations? What does this mean for the utility of a dN/dS approach?
2. I did not follow how the authors estimate that 50% of the total number of CH drivers have now been identified (lines 239-241). Perhaps a supplementary note might further clarify this claim.
3. Could the authors project how growing sample size contributes to the 'completeness' of the CH driver discovery? Eg can they project how many additional drivers would be identified if the entire 450,000 person UK biobank dataset were used or how many samples would be required to identify all drivers.
4. It appears that U2AF1 is largely missing from this analysis. Presumably this is related to the hg38 reference assembly and would be worth either acknowledging in the text as a limitation that is not likely to affect the results or modifying the analytic approach to account for this (eg: https://github.com/weinstockj/pileup_region).
5. I would temper some of the claims around novel genes discovered. CHEK2 is widely used in CHIP analyses (eg ref 36) and as noted in the text SRCAP, ZNF318, ZBTB33, and YLPM1 were identified by Beauchamp (ref 46) and have been incorporated into other published analyses.
6. Figure 2A could probably be an extended data figure.
7. Would check Fig 2C and 2D in reference to the legend. It does not appear to line up to me. It appears that 2C is replotting a subset of 2B instead of fitness estimates for MTA2 (which may be what is plotted in 2D) 2D appears to have a x-axis sub title of "Lorem ipsum"
8. Extended Data Fig 2B&C- in the legend the authors may wish to specify what the Green/Orange/Blue color schemes represent.
9. Several of the preprints included on the reference list (eg ref 24 & 25) have been published for more than a year and should be updated accordingly (or merged with other references on the list eg ref 31).

-Alexander Bick

Author Rebuttal to Initial comments

26th November 2023

Referee expertise:

Referee #1: clinical, hematology oncology

Referee #2: WGS, blood, clonal hematopoiesis

Referee #3: clonal hematopoiesis genetics (signed report)

We thank our Reviewers for their valuable feedback. Please find below responses to each comment. Our reply is in blue, and a description of any changes we have made to our manuscript in red. Corresponding changes in the revised manuscript are also highlighted in red. Reviewer comments below are in black.

Reviewers' Comments:

Reviewer #1:

Remarks to the Author:

The manuscript from Bernstein and Chapman et al. "Pervasive positive selection in blood in 200,618 individuals and novel drivers of clonal hematopoiesis" reports an analysis of the UK Biobank (UKBB) dataset identifying genes that are clonally expanded in the blood of study participants. They present evidence that these genes are under positive clonal selection. They show that CH with mutations in these genes correlates with poor outcomes with similar to what has been established for CH.

The text is well written overall and the figures are nicely presented. While this study certainly has something to contribute to the field, and the findings are of significant interest, the central advance of this paper is the identification of additional CH genes which is more incremental than innovative at this point in time. Additionally, several technical issues should be addressed:

Major points:

(1) Most of the manuscript is built on concepts of dN/dS and selection coefficient (s) borrowed

from evolutionary theory. This is certainly a valuable perspective, however this methodology makes a number of assumptions about the data that may not optimally apply to the data, and as a field we need to be careful about using this measurement/calculation as an arbiter of truth. The authors appropriately acknowledge the situation of somatic hypermutation in the lymphoid compartment which is one example. In addition that some other issues warrant at least discussion as limitations.

(1a) Peripheral blood VAF does not measure VAF in the cells which acquire and maintain mutant clones (such as HSCs) of the blood population. The dynamics and interpretation are therefore fundamentally different than ie a population of poorly differentiated carcinoma cells in which most have high (?unlimited) replicative potential. One consequence of this is that mutations that improve HSC fitness and replication while also blocking differentiation would be interpreted to be deleterious to fitness (fewer mature daughter cells in the peripheral blood) while mutations that increases propagation only in later stages of differentiation and do not actually alter HSC fitness might be interpreted as "drivers". I applaud the author's use of the single cell derived

hematopoietic colony dataset as an orthogonal method of validation which addresses the issue for about half the FI genes, but there are still the others that were not validated and conceptual factors could be at play in addition to the sample size issue discussed.

We thank the reviewer for this interesting point. Whole blood contains a mixture of mature lymphoid and myeloid cells and our VAF measurements reflect the average VAF across this cell population. As discussed in the paper, there are examples of genes (e.g. *MYD88*, *IGLL5*) where we believe - based on information from colonies (Table S9), gene expression (Figure 3C), or pre-existing literature - that the expansion is predominantly within the lymphoid compartment. In such cases, the selection may well be absent at the HSC level, being present instead within a long-lived lymphoid population. As such, an HSC-specific approach may miss such CH drivers.

Outside of such examples, we assume that clonal fractions in peripheral blood are broadly reflective of those in the HSC population. The reviewer points to certain scenarios in which this may not be the case. We will address how we predict these would affect our data, and therefore how the data should be interpreted:

- (1) Mutations that improve HSC fitness while blocking differentiation. These mutations will indeed be more difficult to detect through our approach, which requires a VAF > 0.1 in whole blood. Such mutations must therefore be considered a relative 'blind spot' of our method. Indeed, any method that involves sequencing of the peripheral blood compartment to infer driver mutations that solely promote HSC fitness but lead to no expansion in whole blood, would suffer from the same issues. However, we do not

believe that substantial HSC fitness promoting drivers would be missed for 3 reasons. (i) Many 'drivers' that confer HSC fitness (eg *DNMT3A*) are detectable in peripheral blood due to concomitant expansions of more mature cells. (ii) The drivers that promote HSC expansion whilst more completely blocking differentiation (eg some AML drivers) also lead to spill out of immature cells into the peripheral blood at higher clonal fractions (iii) We do not see any genes displaying evidence of negative selection by dN/dS, which we would expect for mutations which would result in absence in mature cells. However, to rightly acknowledge the concept that haematopoietic cellular compartment specific clonal fractions might not be detectable in whole blood, we have added the following statement to the discussion "Studies of different haematopoietic populations, such as haematopoietic stem cells or specific lymphoid populations not represented in whole blood, may also identify additional genetic sites under positive selection." (line 302, page 10)

- (2) Mutations that increase propagation only in later stages of differentiation and do not actually alter HSC fitness. It is true that a mutation may cause a particular advantage later in differentiation leading to increased VAFs in mature cell fractions. However, when considering the myeloid lineage, if the clone is only present at the progenitor stage or beyond, it will be transient and unlikely to reach high VAF levels, as the progenitor clone will become exhausted. For any clone to be persistently detectable it must at the very least fix in the HSC population (or an equivalent population with long term self-renewal capability), which requires for a selective advantage (eg increased self-renewal rate) at the level of self-renewing stem cells. Fixation in the HSC pool is unlikely for clones with no advantage as the strength of selection must be greater than [HSC generation time/HSC population size] for any clone to exceed the genetic drift threshold and fix within a population¹. Additionally, it is unfeasible for a clone to reach a VAF > 0.1 in peripheral blood

(i.e. a clonal fraction > 0.2) without some expansion at the HSC level. Therefore, while mutations may increase propagation later in differentiation, giving VAFs in the mature compartment greater than those in the HSC compartment, we predict that such clones will still have a selective advantage at the HSC compartment. Overall therefore, we have more sensitivity to detect such clones, but we do not believe that there will be any resulting false positives.

Together, we believe that the scenarios outlined by the reviewer will increase or decrease our sensitivity to detect genes under selection, but will not lead to false positives.

In the context of age-related clonal haematopoiesis at least, extreme examples of these

scenarios appear to be relatively rare. This is based on data from other studies.

- (1) In Mitchell et al (Nature 2022)², HSC-derived colonies did not show significant phylogenetic clustering separate from stem/progenitor-derived colonies, suggesting similar clonal contributions within these populations.
- (2) In Campbell et al (Research Square PREPRINT available at <https://doi.org/10.21203/rs.3.rs-2868644/v1>), there was a high concordance between clonal fractions predicted from single progenitor-derived phylogenies and targeted sequencing of mature myeloid fractions (Rebuttal Figure 1). This suggests at least similar clonal fractions between myeloid progenitors and mature cells.
- (3) In Williams et al (Nature 2022)³ studying the driver mutations of myeloproliferative neoplasms that result in strong proliferative advantages to progenitors and more mature cells, the selection estimates from the phylogenies matched up well with fitness estimates from clonal fractions in peripheral blood, unless there was interferon therapy that had intervened before mature blood cell sampling. Furthermore, phylogenetic trees demonstrated expansion of ancestral HSCs.

Rebuttal Figure 1 taken from Campbell et al preprint 2023. Clonal fractions inferred from myeloid progenitor-based phylogenies compared to targeted sequencing of monocytes. Plot shows only clones that are at least 5% clonal fraction in either transplant donors or recipients. The x-axis shows clonal fractions inferred from the proportion of colonies from that individual coming from that clone, with error bars giving the 95% confidence interval (exact binomial test). The y-axis shows clonal fractions inferred from the deep targeted sequencing of monocyte fractions. Confidence intervals for the targeted sequencing data are generally narrow and therefore not shown.

With regards to why some mutations haven't been validated in the colonies, we suspect that sample size is the biggest factor. The colony data is from within only 50 individuals, though each person has been extensively surveyed across many blood cell genomes. This small number of individuals already validated 11 of the 17 genes (including *IGLL5*, which we believe to be mutated due to somatic hypermutation). Whilst this approach gives good power to find commonly mutated genes (even those that may have a relatively weak selective coefficient), it will not detect rarely mutated genes, even if under strong selection.

We would like to highlight that our VAF distribution analysis from which we can estimate the strength of a mutation's fitness (Figure 3B), is further orthogonal validation of the gene level fitness inferred from dN/dS, which we apply to genes where individual sites are sufficiently recurrently mutated. This completely independent approach using an evolutionary framework to estimate positive selection⁴, and provides orthogonal validation for an additional 2 genes (*SRSF1* Glu60Asp and *MTA2* Asp289Gly/*MTA2* Asp293Gly). Of note, these two genes do have 5 missense mutations collectively within the colony dataset, and no synonymous mutations detected, however, the mutation number is too infrequent for additional validation by dN/dS in the colony dataset.

We would also like to highlight that mutations in the colony dataset were actually found in 16 of the 17 genes, with similar mutation spectrums to UKBB with a total of 150 mutations. Of these, 125 mutations were found within healthy individuals with myeloid or lymphoid colonies (Table S10), confirming that positive selection on these variants is active during normal health. This is important as several of the genes reported thus far have only been identified in individuals with cancers or post therapy⁵⁻⁸.

Excluding *IGLL5*, the significance of which remains to be determined as we discuss in the paper, and *MYD88*, a previously reported driver of lymphoid malignancies, there are only three remaining genes requiring further validation (*CCDC115*, *ZNF234*, *CCL22*). These 3 genes were incidentally the least frequently mutated in our cohort, and as a result may not have validated from the orthogonal methods employed. We did observe rare mutations in *ZNF234* (two variants including one nonsense mutation) and *CCL22* (one variant) present in the colonies. We have added the following to the results section:

(line 203, page 7) "Overall, across >10,000 single cell derived clonal haematopoietic cell colonies derived from 50 individuals, we identified 150 somatic mutations in 16 of the 17 new FI-drivers, with a strikingly similar pattern of non-synonymous mutations to bulk whole exome sequencing data from UKBB. The majority of variants (n=125 of 150) were from myeloid and lymphoid colonies from individuals with healthy haematopoiesis (Table S10), with no history of

malignancy or chemotherapy exposure, demonstrating that positive selection acts on mutations in new FI-CH genes during healthy haematopoiesis. Importantly, 10 of the 17 new FI-CH genes were also under positive selection, as estimated by dN/dS in single-cell derived colonies from a small set of individuals, validating UKBB findings. Two of the remaining seven genes (*SRSF1* and *MTA2*) did show independent evidence of positive selection based on the distribution of the VAFs of recurrently mutated sites (Figure 2B). Excluding *IGLL5*, the significance of which remains to be determined, and *MYD88*, a previously reported driver of lymphoid malignancies, there are three remaining genes (*CCDC115*, *ZNF234*, *CCL22*) identified as under positive selection in UKBB which we could not independently validate. Whilst two *ZNF234* variants were found in haematopoietic colonies, including 1 nonsense mutation, and a missense mutation in *CCL22* was present in a naïve B cell colony, these numbers were too low to display gene wide positive selection. Incidentally,

these three genes were also the least frequently mutated in UKBB, and they should be considered provisional CH genes pending future studies of larger datasets where their potentially increased variant numbers may allow further elucidation.”

(1b) Another way in which human blood systems are different from populations with ideal genetic mixing is that total HSC number (not just diversity) and bone marrow cellularity decrease as humans age. Is it appropriate to apply dN/dS to a population with a decreasing number of individuals and declining habitat?

The dN/dS methodology extrapolates selection solely based on whether the observed number of normalised non-synonymous mutations is more or less than expected relative to the normalised number of synonymous mutations, which are assumed to have no functional significance. Therefore it is agnostic to population dynamics and remains a useful tool in this setting. The decreasing population may however have an impact on the selective pressures resulting from the altered environment that may - in theory - alter the landscape of driver genes through life. This has been previously theorised for mutations in genes such as *TET2* and splicing factor mutations that seem to expand in old age beyond other genes^{9,10}. By applying dN/dS across this period, we are therefore getting a readout of the ‘average’ selection across the UKBB age range that may theoretically dilute results if there were genes with opposing selection at different ages. In reality, we see little evidence of this, with raised global dN/dS ratios in both young and older individuals (Figure 1C). Whilst global dN/dS estimates trend towards a lower ratio in younger individuals (Extended Figure 1C) in this dataset, this reflects the reduced ability of bulk exome sequencing data to capture smaller size clones expected in younger individuals. Importantly, in single cell derived colony datasets where each sample is clonal derived (Mitchell et al, Nature 2022²), the dN/dS ratios are the same in very young to older individuals, confirming

that the rate of entry of driver mutations is constant over life. Therefore, it is appropriate to pool across age ranges to increase our power to detect genes under positive selection.

(1c) How strong is the correlation between dN/dS and # of individuals with a mutation? It looks like a Pearson r would be large based on the dN/dS of genes in figure 1C and number of individuals with mutations in each gene in figure 1H. Which genes could be identified only by this method that we would be blind to if only using number of mutations (obviously we don't need it to see that DNMT3A, TET2, and other highly prevalent genes are under positive selection).

There is indeed a correlation between dN/dS and the number of individuals with a mutation in a particular gene (Rebuttal Figure 2). It is also true that *DNMT3A* and *TET2* are the two most mutated genes as well as being those with the highest dN/dS values. However, beyond this point, the relationship starts to break down for a variety of reasons:

- Large genes (such as *TTN*) have high numbers of mutations merely due to their size
- Even if gene size is taken into account by looking at mutation rate per base, some genes have higher mutation rates than others due to local effects such as chromatin organisation, exon density and GC content¹¹. In such cases, a high mutation rate may be mistaken for positive selection.
- Particular sequence contexts have a high mutation frequency e.g. C>T at methylated CpGs. Genes with a high density of such sequence contexts may consequently have a higher mutation frequency for this reason^{12,13}.

In addition, truncating mutations are rare, but may be strongly selected for. Consequently genes with a specific selection for such mutations may not have an overall significant increase in mutations, though if looking specifically at truncating mutations an increase becomes evident. In this situation, an appreciation of (1) the number of codons that have potential to mutate into stop/ splice site/ frameshift variants, and (2) sequence-specific understanding of the mutation likelihood by chance, is important to interpret if there is true evidence for selection. The *dnscv* package is a helpful way of accounting for most of these potential confounding factors. It combines local information (synonymous mutations in the gene) and global information (variation of the mutation rate across genes, epigenomic covariates) to estimate the background mutation rate, and does this separately for truncating, missense, splice site variants and indels¹³. Therefore, we believe this to be a powerful and appropriate tool to detect selection, particularly for genes where only truncating mutations are selected, as such mutations may be infrequent across individuals despite being under positive selection.

Rebuttal Figure 2. Number of mutations in a gene versus the dN/dS ratio. Size of dots reflects number of variants identified.

(1d) Could there be a systematic issue relating to how dNdScv interacts with variant calling making it so that exonic regions with either poorer read coverage or that tend to be covered near the ends of reads where filters are more stringent are less likely to be called and could bias interpretation ie if they are interpreted as places where the probability of mutation is high but none/few mutations are found?

In general, regions with low coverage will decrease the power to call both synonymous and non-synonymous mutations. Therefore, while this reduces power to detect selection, there should be no systematic impact on dN/dS estimates. However, if there were indeed a mutation hotspot in an area of lower coverage, or in an area with poor variant calling, this impact may be significant, but predominantly because the variants would be hard to detect in order to feed into dNdScv, and not because dNdScv would fail to perform. It is worth noting that dN/dS estimates are done on a 'per gene' basis, and therefore specific technical effects e.g. relating to the ends of exons, will be averaged out across the gene and would be expected to affect synonymous and non-synonymous

mutations equally. Overall, therefore, we believe that while the data may limit our power to detect selection in some circumstances, there should be no significant systematic interaction

with the dNdScv method that would result in false positives.

(1e) I'm surprised that there such an apparently strong MYD88 hotspot mutation is not regarded as strongly positively selected for in the model. The mutation identified is different than L265P which we do clinical testing for in the workup of Waldenstrom's, but is still in a functionally annotated domain that is highly evolutionarily conserved (at least through Danio Rerio). Are there really so many non-synonymous mutation in this region that this hotspot does not look selected for? If so, would a non-coding regulatory element within the DNA sequence who disruption causes synonymous mutations to have a fitness effect be one possible explanation?

The L273P hotspot mutation in *MYD88* in our manuscript, in fact, corresponds to the same L265P hotspot mutation used for diagnostic testing alluded to by the Reviewer. The difference is due to the use of different transcripts for annotation. The L273P mutation in *MYD88* is labelled using the ENST00000417037.6 transcript, the default for many annotation databases including COSMIC (<https://cancer.sanger.ac.uk/cosmic/gene/analysis?ln=MYD88>). However, the ENST00000396334 transcript is also commonly used, resulting in annotation as L265P (https://cancer.sanger.ac.uk/cosmic/gene/analysis?ln=MYD88_ENST00000396334).

To avoid confusion we have clarified this in the manuscript on page 5, line 155 "...MYD88 p.Leu273Pro, (also commonly referred to as Leu265Pro when protein annotation uses transcript ENST00000396334) ..."

(2) There appear to be very few total mutations in some of the 17 FI-CH genes identified by the authors based on figure 2A. For example *CCDC115* appears to be mutated only twice in the entire dataset (or mutated ≈ 2.5 times based on the y-axis labelling). If I am interpreting this correctly, I do not think that 2 mutations is sufficient to calculate a meaningful estimates of the synonymous and non-synonymous mutation rates, or of their. Also related to *CCDC115* it is curious that both of the mutations are so early in the gene because truncating mutations in this position can often be skipped. Does dNdScv treat all truncations the same regardless of position in the gene? This might be a situation in which that is not a good approximation.

We thank the reviewer for this interesting observation. It is true that in a few cases the inference of positive selection is based on a small absolute number of mutations. In the case of *CCDC115*, there are three nonsense mutations within the UKBB dataset. In this example, the significance threshold is still reached due to the following observations:

- The total number of mutations within the gene is extremely low, primarily because it is a short protein of only 191 amino acids. Other than the three nonsense mutations, there

are no other synonymous, missense, or indel mutations. Passenger events are therefore expected to be very rare.

- The overall proportion of truncating mutations by chance is anticipated to be very low (e.g. in TTN, only 5.6% of mutations are truncating, and only 5.3% of *CCDC115* mutations in COSMIC are nonsense mutations). The fact that three nonsense mutations were captured, with no other mutations identified, is therefore highly unlikely by chance, and was found to remain significant even after correction for multiple hypothesis testing.

The Reviewer's observation that the nonsense mutations are all early in the gene is intriguing. *dNdScv* does indeed treat all truncating mutations the same and doesn't account for differences in potential impact based on position within the transcript. Regardless, three identical nonsense mutations in a gene with a low background mutation rate suggests a functional consequence, perhaps indicating that they are not skipped in this case, or else an impact on protein function through other mechanisms.

Notably however, *CCDC115* was the only gene that was not found in the colony-sequencing data (mutations in the other 16 genes were found in the 10000+ colonies). This may be because this is a rare mutation that we were insufficiently powered to detect in the small number of individuals included in the colony data. It is also possible that cells with *CCDC115* mutations were not amongst the types of colonies that were sampled in the colony datasets, or that cells with *CCDC115* mutations do not grow in culture. Taking into account the Reviewer's comment, we have now expanded on our validation strategies used in this paper, and discuss that genes not validated in the colony-sequencing data should be considered provisions until future larger studies can elucidate them further.

(line 214, page 7) "...there are three remaining genes (*CCDC115*, *ZNF234*, *CCL22*) identified as under positive selection in UKBB which we could not independently validate. Whilst two *ZNF234* variants were found in haematopoietic colonies, including 1 nonsense mutation, and a missense mutation in *CCL22* was present in a naïve B cell colony, these numbers were too low to display gene wide positive selection. Incidentally, these three genes were also the least frequently mutated in UKBB, and they should be considered provisional CH genes pending future studies using larger datasets where their potentially increased variant numbers may allow further elucidation."

(3) Several of the "17 new genes" are not new. As appropriately discussed by the authors some commonly mutated genes like CHEK2, SRCAP, and MYD88 are already supported by other literature. This is appropriately cited but it would be helpful if the authors can expand on how their study advances understanding of these genes and more specifically focus on the novel

CH genes they identified.

There have indeed been reports that some of the genes included in this study are driver mutations, and as the Reviewer states, we have taken care to appropriately cite these references, both in the main text and in more detail as a supplementary table (Table S6). This table lists all previous citations including those where such genes were observed as passenger events in sequencing datasets. Table S6 shows that many genes have not been reported to be mutated in blood before with some others described in single reports only. Importantly, several large scale studies on CH still do not include any of these 17 genes.

Nevertheless, we have tempered claims throughout the paper. We no longer refer to ‘novel’ CH genes as these imply that they are all ‘original’ for this study. Instead, we refer to them as ‘additional’ or ‘new fitness-inferred CH’, and define ‘new’ as including both novel genes and those recently reported as CH drivers, in order to distinguish them from the classical canonical genes. Eg abstract (line 17, page 1), where we previously said 17 novel genes, we now say ‘We identify 17 additional genes - *ZBTB33, ZNF318, ZNF234, SPRED2, SH2B3, SRCAP, SIK3, SRSF1, CHEK2, CCDC115, CCL22, BAX, YLPM1, MYD88, MTA2, MAGEC3* - including both novel genes and some recently reported genes not routinely included in CH studies, under strong positive selection at a population

level.’. We have made similar changes throughout the manuscript to appropriately acknowledge previous studies. When using ‘new’ FI-CH as a category in the figures, again, we clearly state that this includes both novel genes and those recently reported, to keep them apart from the classical canonical set of CH drivers.

We feel that our study makes the following advances:

- (1) *Selection landscape*. Some of the genes thus far have evidence of being under selection only in specific settings e.g. *CHEK2* in patients who receive chemotherapy⁵⁻⁷, *MTA2* in patients with pre-leukemia⁸, etc. Our analysis has found evidence of positive selection in an unselected population, which we have validated in colony data from individuals that have not been exposed to chemotherapy or without the conditions in which these genes were previously identified, confirming that their selective advantage is not limited to previously identified contexts but occurs in health.
- (2) *Clinical associations*. We provide hazards for different clinical outcomes from infection to cancer at both the gene level and as a group, which has not been addressed before for genes that have been recently reported, suggesting that the presence of clonal expansions associated with these genes are of clinical consequence.

- (3) Powerful methodology with orthogonal validation. Studies such as Beauchamp et al.¹⁴ have used other methods to identify putative CH drivers e.g. high numbers of nonsense mutations or of passenger mutations at similar VAFs, however, most genes have not been validated in independent study, and such genes are yet to be adopted into large studies of CH. For example, a recent large scale study of 200K UKBB participants from Kar et al, Nature Genetics 2023¹⁵ only includes *MYD88* from the 17 additional genes despite some of the genes recently reported. Our identification here by an independent method such as dN/dS with additional validation in single-cell derived colonies together with estimation of variant specific fitness effects - approaches that control for several potential confounders - considerably strengthen the evidence base for these new CH drivers.
- (4) Independent large UK population. Other genes (e.g. *SRCAP*, *YLPM1*, *ZBTB33*, and *ZNF318*) have been proposed as CH drivers in only one or two studies within single cohorts¹⁴. Therefore, this independent UK based population analysis provides important validation and shows that the presence of these expansions is associated with poorer clinical outcomes in the normal population.
- (5) Fitness estimates. We show fitness estimates for several new/recent CH genes – *MTA*, *SRSF1* and *SPRED2*, showing annual growth rates of clones/year comparable to classical CH drivers such as *DNMT3A* R882H and *ASXL1*. Together, we hope that our study makes a strong case for the routine inclusion of these genes in future CH studies.

We now add (line 206 page 7), “The majority of variants (n=125 of 150) were from myeloid and lymphoid colonies from individuals with healthy haematopoiesis (Table S10), with no history of malignancy or chemotherapy exposure, demonstrating that positive selection acts on mutations in new FI-CH genes during health.” We also add (Line 290, page 10) “Whilst some of these genes have been recently reported as candidate drivers, or identified in the context of concurrent malignancies and therapy, here we show that positive selection on these genes is detectable in unselected populations, within individuals with healthy myeloid and/or lymphoid haematopoiesis, and that individuals who carry a non-synonymous mutation in these genes (at VAF >0.1) having significantly increased hazards for a wide range of adverse health outcomes.”

(4) It wasn't clear to me that an appropriate correction for multiple hypothesis testing was performed in the analysis and interpretation of hazard ratios in figure 3A,B,D,F.

Thank you for highlighting this and our apologies for omitting to show these data. We have now added new Supplementary Tables 12 and 13 that detail the specific hazard coefficients for the clinical outcomes for the different classes of CH (Table S12 corresponding to Figure 3A, B, D, F) and for the different genes (Table S13, corresponding to Extended data 3). These show the adjusted p- values (FDR corrected), and we have directed the reader to these tables within the

main paper. We have also included an asterisk on Figures 3A, B, D and F for associations that are significant after FDR correction (<0.05).

Minor points:

(1) Abstract text lines 14 and 15: The statement that CH is typically driven by somatic mutations in a small set of genes but that most clonal expansions have unknown drivers seems self-contradicting, or at least isn't easy to understand.

We agree that this wording is confusing, and have therefore updated it as follows (page 1, line 14) "CH is frequently driven by somatic mutations in a small set of canonical genes, however the majority of clonal expansions detectable in blood lack known drivers."

(2) Figures 1A, 1E, and 3E, and extended data Figure 3 show model fits of data. This is very pretty but please show the actual data somehow like for example median and std deviation in 2 year bins. The model used to fit the data and meaning of error bars (nearly invisible in my printed copy in the main figure panels) should be described in the figure legend.

Thank you for this comment. We have now changed the following figures (Fig 1A, 1E, 1F, 3E and Extended Figure 3) plots to show the actual data in bins, together with error bars representing 2*standard error behind the model fits of the data. We have also added to the legend that the fit used was a second degree polynomial. We have added the explanation of what has been plotted to the legends as follows, (line 319 page 11 and line 354 page 12) "Plots show the data in 2 year bins together with error bars representing 2*standard error. Smoothed line together with shading represents a second degree polynomial fit of the actual data."

(3) Main text line 96: "infrequent mutation infrequency" is redundant.

Thank you for this observation. This sentence has now been updated.

(4) At the bottom of Figure 2D the words "Lorem Ipsum" are printed apparently in error.

Thank you, this has been removed.

(5) The way in which the supplementary tables are enumerated was garbled at least in the combined PDF version I received.

We apologise for this. We have checked our original versions uploaded and the error seems to

be in the conversion step online used to generate the pdf. Whilst we can still see the excel spreadsheet

version online which is showing no issues, we are unsure if this format was made visible to Reviewers. We will ensure that we ask the Editorial team to make available the excel spreadsheet format of the supplementary tables at revision without conversion.

(6) In the interest of transparency and to protect the authors from the appearance of impropriety then if any authors have stock and/or ownership stake in Calico and/or parent company Alphabet that should be disclosed in addition to the employment relationship. If Calico and/or any authors have filed provisional patents to protect intellectual property related to this manuscript that should be disclosed as well.

There are no IP filings from Calico or the Wellcome Sanger Institute. We can confirm that none of the authors not employed by Calico (ie Wellcome Sanger coauthors) have any financial relationship with Calico or Alphabet, specifically, there are no stocks/ownership stake, and no funding has been received by any authors, such as in the form of research grants or honoraria.

Reviewer #2:

Remarks to the Author:

-In this manuscript, the authors use data from >200k participants in the UKBiobank with whole-exome sequencing (WES) data to infer gene-level selection in the context of clonal hematopoiesis. They identify 17 new genes that are the targets of positive selection. Single-cell WGS data were used to verify that these 17 genes are under positive selection and to identify the stem cells/progenitor cells in which these mutations are present. In so doing, they define a set of “non-canonical fitness-inferred” CH driver mutations that have fitness effects comparable to canonical CH drivers and that also increase in frequency and size with age. This study represents a clever use of the UKBB WES data and dN/dS ratios to expand the set of known CH driver mutations. Given this expanded set of CH drivers, they estimate that “driverless” CH represent about half of all CH cases. As far as I could tell, the study contained no real conceptual or technical flaws. The paper was also very well-written. Therefore, the following are a list of suggestions to strengthen what is already an excellent paper.

In the Introduction it is worth mentioning that the prevalence of clonal hematopoiesis (CH) is highly dependent on what mutant cell fractions (CF) are identifiable. For instance, defining CH as clones with CF > 5% leads to a much different estimated prevalence than if CH is defined as a clone with CF > 0.1%.

We thank the Reviewer for this important point. We have now added the following statement

to the introduction, (page 2, line 38) “However, the prevalence of CH is highly dependent on the sensitivity of sequencing assays, with very small CH clones reported in most individuals over the age of 50 when using highly sensitive sequencing.” citing Young et al¹⁶.

-Can the authors comment on how the disease status of the 50 individuals with sc-HSC sequencing data may have affected their results?

We thank the reviewer for raising this point. For the most part, we do not believe disease status to have had a major impact on our findings. First, a large number of individuals in the cohort had no known blood disease (n=29 of 50, individuals from post-transplant, clonal haematopoiesis, ageing and foetal studies). These also tended to be those with larger numbers of samples, such that 83% of whole genomes (8792 of 10635) were from these individuals. Similarly, 80% of

mutations in the 17 novel FI-CH genes were from these individuals. Within these ‘normal’ groups, there may be some distinctive features:

- The three individuals with CH were selected because of known expansions in canonical CH mutations. Theoretically this may decrease the power to detect other drivers, as the fittest driver clone may have outcompeted weaker drivers. Conversely, it may result in the detection of mutations that are synergistic with the dominant driver. However, we believe this to be a weak effect as only 3 of 51 individuals, and 7 of 128 mutations were from the CH cohort.
- Within the transplant group, half came from recipients. HSCs in these individuals have been through the unusual situation of being harvested, stored ex vivo, and transplanted into a new individual. Again, this distinctive selective environment may enhance positive selection on some of the genes such as *CHEK2*, *SRCAP*, *ZNF318*, that were mutated in 4 colonies each from this cohort. Although, mutations in these 3 genes were more prevalent in healthy individuals (both normal ageing and transplant donors).

However, amongst the 17 novel FI-CH genes, there was no distinct pattern of mutations in these individuals compared to the normal individuals.

The MPN group had large dominant clones with mutations in canonical drivers, decreasing the power to detect other drivers unless they synergistically interact with the dominant mutation. In this context, *SRCAP* appeared to be relatively enriched (16% [5/31] *SRCAP* mutations, compared to 9% [11/128] of all mutations in 17 genes).

The tAML sample was relatively enriched in novel FI-CH mutations, with 8% (10/128) mutations

found in this individual, despite constituting only 1% of the total colonies. These ten included 3 CHEK2 mutations and three YLPM1 mutations. CHEK2 has been described as being under particular selection with chemotherapy (Bolton et al, Nature Genetics, 2020), and this suggests this may also be the case for YLPM1.

To clarify the distribution of mutations between the different datasets, we have added this summary table to the manuscript as Supplementary Table 10 also shown below and referenced it in the manuscript. We feel the key message is that the majority of the mutations validated in colonies from individuals with normal haematopoiesis.

Dataset	Number of individuals	Number of colonies	BAX	CCL22	CHEK2	IGLL5	MAGE3	MTA2	MYD88	SH2B3	SIK3	SPRED2	SRCAP	SRSF1	YLPM1	ZBTB33	ZNF234	ZNF318
Normal ageing	12	4669	1	0	8	0	3	1	1	1	5	2	22	1	7	3	0	15
Therapy-related AML	1	99	1	0	3	0	0	0	0	0	0	1	0	0	3	1	0	1
Clonal Haematopoiesis	3	267	0	0	3	0	1	0	0	0	2	0	0	0	0	0	0	1
Chronic myeloid leukemia	7	580	0	0	1	0	0	0	0	0	0	0	0	0	0	0	0	1
Foetal	2	511	0	0	0	0	0	0	0	0	0	0	0	0	0	0	0	0
Allogeneic HCT	10	2912	0	0	4	1	1	1	0	3	1	2	4	0	0	3	1	4
Myeloproliferative Neoplasms	12	1013	1	0	1	0	1	1	0	1	0	0	5	0	0	1	0	0
Triple-negative ET	3	151	0	0	1	0	0	0	0	0	0	0	0	0	0	0	1	0
Lymphoid	7	735	0	1	1	15	1	1	0	1	0	1	0	0	2	0	0	0

-An additional analysis that may be of interest would be to compare clonal growth rates (estimated via PACER, PMID: 37046083) between the group of individuals with canonical CH drivers and the group of individuals with non-canonical-FI CH drivers.

While we agree that understanding clonal growth rates would be of interest, we suspect that PACER would not be a useful tool for our data. PACER was developed for whole genome analyses and has not been validated for exome sequencing. Given the ~1% of the genome covered by exome sequencing, we anticipate the resolution to be insufficient to give useful estimates of mutation timing. We also have more general concerns regarding the approach of PACER - which assumes that all passenger mutations belong to the same driver-containing clone - when work has shown that blood frequently becomes oligoclonal in old age² such that detected mutations, both passengers and drivers, may be spread amongst several different clones that may be of similar size. Given these clones are often of similar sizes, it is challenging to differentiate which clones passenger mutations might belong too in bulk whole genome sequencing, and even more so in exome sequencing. Thus, growth estimates for mutations found more frequently in elderly populations (e.g. *TET2*, *SRSF2*) will be confounded by the presence of independent clonal expansions, each with its own passenger mutations. Given that our data are exome sequencing, and the need for a license to use PACER, we have not applied it to our data.

Can the authors comment on the lack of a statistically significant association between the novel FI-CH and MI-Stroke (in comparison to the Canonical FI-CH mutations)? Is this just a matter of statistical power (i.e., are there many fewer carriers of novel FI-CH mutations than carriers of canonical FI-CH mutations)?

The point estimate for hazard ratio of New FI-CH and classical FI-CH respectively with MI-Stroke incidence is almost identical, indicating this is likely an issue with power (Rebuttal Figure 3). We demonstrate at an effect size of 1.25 we only have a ~55 percent chance at detecting a significant association of New FI-CH with MI-Stroke incidence. Furthermore, given that some of our new FI-CH genes are lymphoid drivers, we are not concerned about a lack of MI-Stroke comparison. A recent paper does show that the correlation with MI-Stroke is not found when CH categories are broader, such as when including mutated genes selected within lymphoid populations (Stacey et al, Nature Genetics Nov 2023¹⁷).

Rebuttal Figure 3. Power analysis for detecting association with MI-stroke and types of CH. Novel FI-CH refers both to novel and recently reported CH genes. Canonical genes refer to classical CH genes as detailed in methods.

Reviewer #3:

Remarks to the Author:

In “Pervasive positive selection in blood in 200,618 individuals and novel drivers of clonal haematopoiesis”, Bernstein and colleagues infer gene-level selection in blood using exome data from the 200k exome release of the UK Biobank. They identify a set of genes under positive selection with similar selection characteristics to the set of widely established CH driver genes.

Overall, the analysis is very robust and thoughtfully executed. Several minor aspects of the presentation could use additional attention.

1. It is surprising that based on the fitness effect estimated by the dN/dS method here, DNMT3A R882 is among the most fit mutation, however in prior work co-authored both by this group (eg Fabre, Nature, 2022, Ref 24) and observed in multiple other studies (eg Robertson, Nature Med 2022, Weinstock, Nature 2023), DNMT3A R882 is among the least fit. How do the authors reconcile these observations? What does this mean for the utility of a dN/dS approach?

Our fitness estimates in Figure 3 do not in fact make use of the dN/dS values, but only the VAF distributions of recurrently mutated sites (where $n \geq 20$) in the same way as previous models (Watson et al, 2020)⁴. The reason for this is that dN/dS approaches do not account for VAF, or more generally clone size. As long as a clone reaches the limit of detection in the assay they are treated the same, so highly frequent but small clones can have outsized dN/dS ratios, whereas infrequently mutated but strongly selected mutations may have lower dN/dS ratios. There has been some effort in the field to try to combine these dN/dS methods with inferred fitness parameters to get a more holistic view of the fitness landscape but it has only been applied to the solid tissue setting¹⁸. Indeed, this particular method would not work well in a setting in which our limit of detection is quite low, e.g. UKBB low-depth whole-exome sequencing.

Comparing our fitness estimates to those from other studies, DNMT3A R882H was inferred to be one of the most fit by the method described in Watson et al. 2020⁴. In that paper, they saw DNMT3A R882(C/H) to be among the most fit point mutations. The variation in estimated fitness coefficients from our paper to their paper is well within the estimated confidence intervals reported. The very slight difference in point estimates between the two studies may be due to the different UKBB cohorts studied and variant calling strategies. Similarly, in Robertson et al, 2022¹⁹, their estimates for the fitness of R882H of ~16% is also similar.

The lower estimates in Fabre et al⁹ of ~5% is indeed interesting. However, we believe that this may relate to several differences. First, this study undertook deep sequencing down to 1000x depth, giving insight into many more low level clones across genes associated with CH. Such small clones would not have been captured in our study. The mean fitness effect for DNMT3A R882H in Fabre et al⁹ was based on six clones, five of which reached a maximum VAF of <0.05, with only one clone reaching a VAF of 0.34. Indeed, the individual with the larger clone had a fitness estimate in keeping with those reported by other studies^{4,19} and our study. The other five clones that remained at very low levels during the longitudinal capture, may represent clones that behave differently, either due to their timing of onset of clonal expansion, or the elderly

nature of the cohort. That study did propose that *DNMT3A* mutations may be preferentially expanding early in life, with a weaker advantage in old age⁹. Thus, our fitness estimates, which are restricted to larger clones that would be detectable by shallower whole-exome sequencing, do indeed have quite consistent fitness estimates across studies.

At the level of dN/dS, one can ascertain at a gene level that *DNMT3A* mutations are under strong selection due to the very high dN/dS ratios (Figure 1C) shown by the gene.

2. I did not follow how the authors estimate that 50% of the total number of CH drivers have now been identified (lines 239-241). Perhaps a supplementary note might further clarify this claim.

We agree with the Reviewer that this estimate could benefit from greater clarity, and have therefore added what follows below as a Supplementary note on page 13, line 389.

Supplementary Note 1

Our estimate of the proportion of driver mutations that have been identified follows the same logic as that used in Mitchell et al². First, we estimate the total number of driver mutations within the set of coding mutations fed into the dndscv algorithm. This is done using the 'global dN/dS' measure i.e. a measure of the total excess of non-synonymous mutations above that expected from the number of synonymous mutations. In our case the global dN/dS is 1.13 (95% CI 1.11 - 1.16). This implies that there are an additional 13 non-synonymous mutations for every one synonymous mutation beyond that expected (ie following correction for the numbers of non- synonymous and synonymous sites respectively), which are assumed to be those under selection. The absolute number of non-synonymous drivers, N_{DRIVERS} , can be calculated from the total number of non-synonymous mutations in the set, N_{ACTUAL} , as follows: $N_{\text{ACTUAL}}/N_{\text{PREDICTED}} = 1.13$. But N_{ACTUAL} is a combination of non-synonymous mutations occurring by chance (which will be the same as $N_{\text{PREDICTED}}$) and non-synonymous mutations present due to selection (N_{DRIVERS}). Therefore: $(N_{\text{PREDICTED}} + N_{\text{DRIVERS}})/N_{\text{PREDICTED}} = 1 + (N_{\text{DRIVERS}}/N_{\text{PREDICTED}}) = 1.13$

Rearranging this gives $N_{\text{DRIVERS}}/N_{\text{PREDICTED}} = 0.13$, ie., $N_{\text{DRIVERS}} = 0.13 * N_{\text{PREDICTED}}$

But from the original equation, $N_{\text{PREDICTED}}$ is $N_{\text{ACTUAL}}/1.13$, therefore, $N_{\text{DRIVERS}} = (0.13 \times N_{\text{ACTUAL}})/1.13$

In our case, N_{ACTUAL} is 39,083 and therefore N_{DRIVERS} is calculated as 4,496 (95% CI 3,873 - 5,333). Importantly, this global figure includes drivers that we know about i.e. those in driver genes that we have identified, and those that we have not yet identified. Therefore, we next look at the total number of mutations within our new set of CH driver genes within this set of 39,083 non-synonymous mutations. In our case this is 2,270. Finally, we calculate the proportion of the total

set of driver mutations that are now identified which is $2,270/4,496 = 0.5049 = 50\%$ (95% CI 43 - 59%). The 95% confidence intervals are calculated in exactly the same way as the median values, but using the lower and upper bounds of the original dN/dS values i.e. 1.11 - 1.16.

3. Could the authors project how growing sample size contributes to the ‘completeness’ of the CH driver discovery? Eg can they project how many additional drivers would be identified if the entire 450,000 person UK biobank dataset were used or how many samples would be required to identify all drivers.

Thank you for this interesting question. To explore this idea, we down sampled our cohort to a variety of fractions of the total cohort size, and reran `dndscv` on the reduced set of driver mutations identified to be significantly under positive selection ($q < 0.1$). We find that for the first 200k samples, the number of genes reaching significance for positive selection for a given sample size is linear (Rebuttal Figure 4). Whether this linear relationship would hold for the full additional 250k individuals is unknowable without strong assumptions about the types of mutations we may identify in a larger cohort, but increasing sample size shows no signs of asymptotic returns in driver discovery thus far. This would suggest that there is a tail of infrequently mutated genes that could

reach significance thresholds in larger cohorts were their mutation numbers to be larger. Therefore, we propose that there would be merit in a much larger future analysis, combining UKBB and other large non-UK cohorts (eg Topmed, DeCODE, etc), to both validate the novel genes identified here, and potentially identify further genes under positive selection in blood.

Rebuttal Figure 4. Dndscv linearly recovers more genes under positive selection with increasing sample size. Each blue dot represents the number of genes reaching significance thresholds ($q < 0.1$, y axis) at a particular fraction size (x axis).

We have added a Supplementary note 2 with this discussion and Rebuttal Figure 4 above (line 414 page 14 of the manuscript). We have also added the following to the end of the Results section (line 270 page 8), “Since *dNdScv* appears to linearly recover increased numbers of drivers of CH within this cohort (Supplementary note 2), there would be merit in a future substantially larger study combining genome sequencing of several different population cohorts to identify further genes under positive selection in blood.”

4. It appears that U2AF1 is largely missing from this analysis. Presumably this is related to the hg38 reference assembly and would be worth either acknowledging in the text as a limitation that is not likely to affect the results or modifying the analytic approach to account for this (eg: https://github.com/weinstockj/pileup_region [github.com]).

We thank the reviewer for this important observation. This is indeed a limitation of the hg38 reference assembly that we should acknowledge. We have added the following text to both results (page 3 line 85) “U2AF1 is missing from this set due to recognized issues with mutation calling relating to the hg38 reference assembly genome²⁰.” We also acknowledge this in the methods section ‘Somatic variant calling in UKBB and *dNdScv* analysis’ on page 13 of the manuscript.

5. I would temper some of the claims around novel genes discovered. CHEK2 is widely used in CHIP analyses (eg ref 36) and as noted in the text SRCAP, ZNF318, ZBTB33, and YLPM1 were identified by Beauchamp (ref 46) and have been incorporated into other published analyses.

We have taken this point fully on board. We no longer refer to ‘novel’ CH genes in the entire manuscript apart from the title of the paper, to avoid the suggestion that all the genes are ‘original’ findings in this study. Instead, we refer to them as ‘additional’ or ‘new fitness-inferred CH’ and

define ‘new’ as including both novel genes and those recently reported, in order to distinguish them from the classical canonical genes. For example, in the abstract (line 17, page 1), where we previously said 17 novel genes, we now say ‘We identify 17 additional genes - *ZBTB33*, *ZNF318*, *ZNF234*, *SPRED2*, *SH2B3*, *SRCAP*, *SIK3*, *SRSF1*, *CHEK2*, *CCDC115*, *CCL22*, *BAX*, *YLPM1*, *MYD88*, *MTA2*,

MAGEC3 - including both novel genes and some recently reported genes, under strong positive

selection at a population level. These additional genes...'. We have made similar changes throughout the manuscript, figures and legends as highlighted, to provide a balanced report. For example, Page 4, line 95 "We called the remaining set of 17 genes "new FI-drivers" of CH, to distinguish them from the classical set of CH genes, as shown in Fig.1C-D (Table S6). These genes

- *BAX, CCL22, CCDC115, CHEK2, IGLL5, SH2B3, SIK3, SPRED2, SRCAP, SRF51, MAGEC3, MTA2,*

MYD88, YLPM1, ZBTB33, ZNF234 and ZNF318 - included novel genes, recently reported candidate drivers of CH in independent datasets, and some previously reported in association with malignancy (Table S6). " On page 7, line 203, we now discuss the validation strategy in detail and suggest that *ZNF234, CCL22 and CCDC115* should be considered as provisional CH drivers pending larger studies.

6. Figure 2A could probably be an extended data figure.

Thanks for the suggestion. However, we'd prefer to leave this as a main figure because it is the only figure in the manuscript that conveys the intragenic location of mutations affecting "novel" fitness-inferred genes, as well as the nature of those variants. Given that there is spatial clustering of mutations in some genes (*CCDC115, IGLL5, YLPM1* etc) and a difference between which genes are affected by nonsense/indel mutations versus those predominantly affected by missense mutations, we have kept the figure unchanged for now. This figure was of interest to Reviewer 1 due to the clustering of mutations in *CCDC115*, and we feel may be studied by readers.

7. Would check Fig 2C and 2D in reference to the legend. It does not appear to line up to me. It appears that 2C is replotting a subset of 2B instead of fitness estimates for MTA2 (which may be what is plotted in 2D) 2D appears to have a x-axis sub title of "Lorem ipsum"

We apologise for the confusion due to our error. The legend for Fig 2C is actually referring to 2D, and the legend for 2D is referring to 2C. Figure 2C is showing the recurrent mutations with the *highest estimated mutation rates* (as opposed to 2B which is showing those mutations with the *highest estimated fitness effects*). We have now corrected the legend numbering in the text, and also removed the 'lorem ipsum'.

8. Extended Data Fig 2B&C- in the legend the authors may wish to specify what the Green/Orange/Blue color schemes represent.

Thank you. We have now added a key to Extended data Figures 2 and 3 as well as other figures where this was missing.

9. Several of the preprints included on the reference list (eg ref 24 & 25) have been published for more than a year and should be updated accordingly (or merged with other references on the list eg ref 31).

We apologise for the oversight which was due to duplicate entries for these publications in our reference manager. We have now corrected these. The only remaining preprint reference is now

ref 31 (Benjamin et al. Calling Somatic SNVs and Indels with Mutect2) which, to our knowledge, remains in bioRxiv.

-Alexander Bick

References

1. Koonin, E. V. Splendor and misery of adaptation, or the importance of neutral null for understanding evolution. *BMC Biology* 14, 114 (2016).
2. Mitchell, E. *et al.* Clonal dynamics of haematopoiesis across the human lifespan. *Nature* 606, 343–350 (2022).
3. Williams, N. *et al.* Life histories of myeloproliferative neoplasms inferred from phylogenies. *Nature* 602, 162–168 (2022).
4. Watson, C. J. *et al.* The evolutionary dynamics and fitness landscape of clonal hematopoiesis. *Science* 367, 1449–1454 (2020).
5. Bolton, K. L. *et al.* Cancer therapy shapes the fitness landscape of clonal hematopoiesis. *Nat Genet* 52, 1219–1226 (2020).
6. Pich, O., Reyes-Salazar, I., Gonzalez-Perez, A. & Lopez-Bigas, N. Discovering the drivers of clonal hematopoiesis. *Nat Commun* 13, 4267 (2022).
7. Slavin, T. P. *et al.* Association between Clonal Hematopoiesis and Late Nonrelapse Mortality after Autologous Hematopoietic Cell Transplantation. *Biol Blood Marrow Transplant* 25, 2517–2521 (2019).
8. Corces-Zimmerman, M. R., Hong, W.-J., Weissman, I. L., Medeiros, B. C. & Majeti, R. Preleukemic mutations in human acute myeloid leukemia affect epigenetic regulators and persist in remission. *Proc Natl Acad Sci U S A* 111, 2548–2553 (2014).
9. Fabre, M. A. *et al.* The longitudinal dynamics and natural history of clonal haematopoiesis. *Nature* 606, 335–342 (2022).

10. Abelson, S. *et al.* Prediction of acute myeloid leukaemia risk in healthy individuals. *Nature* 559, 400–404 (2018).
11. Hodgkinson, A. & Eyre-Walker, A. Variation in the mutation rate across mammalian genomes. *Nat Rev Genet* 12, 756–766 (2011).
12. Greenman, C., Wooster, R., Futreal, P. A., Stratton, M. R. & Easton, D. F. Statistical analysis of pathogenicity of somatic mutations in cancer. *Genetics* 173, 2187–2198 (2006).
13. Martincorena, I. *et al.* Universal Patterns of Selection in Cancer and Somatic Tissues. *Cell* 171, 1029–1041.e21 (2017).
14. Beauchamp, E. M. *et al.* ZBTB33 Is Mutated in Clonal Hematopoiesis and Myelodysplastic Syndromes and Impacts RNA Splicing. *Blood Cancer Discov* 2, 500–517 (2021).
15. Kar, S. P. *et al.* Genome-wide analyses of 200,453 individuals yield new insights into the causes and consequences of clonal hematopoiesis. *Nat Genet* 54, 1155–1166 (2022).
16. Young, A. L., Challen, G. A., Birmann, B. M. & Druley, T. E. Clonal haematopoiesis harbouring AML-associated mutations is ubiquitous in healthy adults. *Nature Communications* 7, 12484 (2016).
17. Stacey, S. N. *et al.* Genetics and epidemiology of mutational barcode-defined clonal hematopoiesis. *Nat Genet* 1–11 (2023) doi:10.1038/s41588-023-01555-z.
18. Williams, M. J. *et al.* Measuring the distribution of fitness effects in somatic evolution by combining clonal dynamics with dN/dS ratios. *Elife* 9, e48714 (2020).
19. Robertson, N. A. *et al.* Longitudinal dynamics of clonal hematopoiesis identifies gene-specific fitness effects. *Nat Med* 28, 1439–1446 (2022).
20. Miller, C. A. *et al.* Failure to Detect Mutations in U2AF1 due to Changes in the GRCh38 Reference Sequence. *J Mol Diagn* 24, 219–223 (2022).

Decision Letter, first revision:

11th Jan 2024

Dear Dr. Nangalia,

Thank you for submitting your revised manuscript "Pervasive positive selection in blood in 200,618 individuals and novel drivers of clonal haematopoiesis" (NG-A63053R). It has now been seen by the original referees and their comments are below. The reviewers find that the paper has improved in revision, and therefore we'll be happy in principle to publish it in Nature Genetics, pending minor

revisions to comply with our editorial and formatting guidelines.

Congratulations!

Best wishes,
Chiara

Chiara Anania, PhD
Associate Editor
Nature Genetics
<https://orcid.org/0000-0003-1549-4157>

Reviewer #1 (Remarks to the Author):

I appreciate the effort of the authors to respond to the comments and revise the manuscript. The technical concerns have been appropriately addressed. Unfortunately, I am not convinced that this manuscript moves the field forward in a significant way. The following major issues remain:

(1) Most of the genes being discussed have previously been identified (SRCAP, CHEK2, MYD88, MTA2, ZNF318, ZBTB33, YLPM1, CCL2, IGLL5, CHEK2, BAX, SH2B3). In my opinion it is really splitting hairs to argue that calculating a high dN/dS for these genes is an advance. The mutations in genes not already described in prior papers (SIK3, SPRED2, ZNF234) are extremely rare.

(2) The correlations between CH and clinical disease are not novel or interesting at least to me. Association of CH with heme malignancy was recently characterized in depth in Weeks et al 2023 NEJM Evidence and Gu et al 2023 Nature Genetics. Association of CH with disease other than heme malignancy were weak and also mostly presented in Stacey et al 2023 Nature Genetics. Furthermore, correlations between CH and clinical outcomes do NOT imply CH has a causal role ("clinical consequence") and should be interpreted cautiously.

(3) DNMT3A has the highest dN/dS but is consistently among the weakest CH associations with heme malignancy. What, if anything, is the practical utility of having a fitness estimate for mutations in a particular gene?

At least half a dozen analyses of 100k+ participant UK Biobank cohorts have already been published. While this particular analysis is technically well done, the novelty of the findings is limited, and I don't think Nature Genetics is the appropriate venue for its publication.

Reviewer #2 (Remarks to the Author):

The authors have done an excellent job of responding to the first round of reviews. I have not further comments.

Reviewer #3 (Remarks to the Author):

I appreciate the author's thoughtful revision. I have no further comments.

Author Rebuttal, first revision:

7th February 2024

Point by point response.

Please find below our responses to remaining points. Our responses are in black, and Reviewers' comments are in blue text.

Reviewer #1:

Remarks to the Author:

I appreciate the effort of the authors to respond to the comments and revise the manuscript. The technical concerns have been appropriately addressed. Unfortunately, I am not convinced that this manuscript moves the field forward in a significant way. The following major issues remain:

(1) Most of the genes being discussed have previously been identified (SRCAP, CHEK2, MYD88, MTA2, ZNF318, ZBTB33, YLPM1, CCL2, IGLL5, CHEK2, BAX, SH2B3). In my opinion it is really splitting hairs to argue that calculating a high dN/dS for these genes is an advance. The mutations in genes not already described in prior papers (SIK3, SPRED2, ZNF234) are extremely rare.

We do not attempt to claim that all these genes are completely novel and openly discuss prior publications which we detail both in the manuscript and in Supplementary Table 6.

Prior reports of some of these genes are generally in a specific context, for example, known haematological malignancy, or as candidate drivers of clonal haematopoiesis in a single previous report.

To summarise,

- (i) *CCDC115*, *SIK3*, *SPRED2*, *SRSF1* and *ZNF234* have not been reported before in the literature as candidate drivers of CH and were identified as under positive selection using dN/dS in over 200K individuals without reported haematological diagnoses from UK Biobank. Of these, *SPRED2* and *SIK3* showed independent evidence of positive selection in haematopoietic colonies derived from single haematopoietic stem or progenitor cells providing orthogonal validation. Furthermore, *SRSF1* and *SPRED* hotspots also showed evidence of positive selection based on the distribution of their variant allele fractions in UK Biobank, again providing further validation for their commonly mutated sites. Mutations in *ZNF234* and *CCDC115* were also identified in haematopoietic colonies but under the threshold for significance. In our opinion, it is important that these genes are reported and recognised as new drivers of CH, so that they can be included in all future CH genomic analyses. Furthermore, they would be interesting candidates for functional analyses.
- (ii) Whilst mutations in *CCL22*, *MAGEC3*, *MTA2* and *MYD88* have been reported in the context of haematological malignancy, apart from *MYD88*, the remaining genes are not included in CH panels or recognised as drivers of clonal expansions in the absence of malignancy. Again, as screening for CH commences in many countries, these genes should be routinely added, particularly as they also represent drivers of lymphoid expansions (*CCL22*, *MAGEC3* and *MYD88* in particular).
- (iii) *BAX*, *CHEK2*, *SH2B3*, were reported by Pich et al as new candidate drivers of CH, *SRCAP* has been reported by Beauchamp et al 2021 and Slavin et al 2019, and *YLPM1*, *ZBTB33* and *ZNF318* have been reported as candidate CH drivers by Beauchamp et al 2021. Here we bring together their identification in a large population cohort that we believe more comprehensively identifies genes under positive selection, thus validating their role in CH.
- (iv) Previous reports have not assessed the clinical associations of these additional genes which we do in this report for the first time in detail, together with estimates of their fitness effects.

While mutations in *SIK3*, *ZNF234* and *CCDC115* are indeed relatively rare, we believe this will be the case for most remaining CH genes, and nevertheless represent important advances towards identifying the 'complete' driver landscape in CH. Our data suggest that ~50% of the driver landscape in HSCs remains unidentified. Remaining unidentified drivers are likely a 'long tail' of many genes that only rarely give rise to advantageous mutations.

(2) The correlations between CH and clinical disease are not novel or interesting at least to me. Association of CH with heme malignancy was recently characterized in depth in Weeks et al 2023 NEJM Evidence and Gu et al 2023 Nature Genetics. Association of CH with disease other than heme malignancy were weak and also mostly presented in Stacey et al 2023 Nature Genetics. Furthermore, correlations between CH and clinical outcomes do NOT imply CH has a causal role ("clinical consequence") and should be interpreted cautiously.

We do not claim any causal implications between CH and the observed associated clinical outcomes but do believe that the clinical associations reported here are of significant interest to the clinical community. Knowledge of these additional genes and their clinical disease associations is particularly important to explore further in the future as we venture into an era of earlier screening for CH, attempts at risk stratification of patients, and to aid the design of clinical trials to test therapeutic interventions for high risk patients.

(3) DNMT3A has the highest dN/dS but is consistently among the weakest CH associations with heme malignancy.

What, if anything, is the practical utility of having a fitness estimate for mutations in a particular gene?

A fitness estimate provides several biological insights. First, it is a direct read out of how strongly a particular mutation confers a functional advantage to a cell. Whether this functional advantage impacts on subsequent haematological malignancy would depend on the nature of that functional change. For example, *NOTCH1* mutations have a fitness advantage in oesophageal epithelium but protect against cancer, as these clonal expansions impinge on the growth of competing *TP53* mutant clones. Thus, understanding how a particular mutation provides less or more of a fitness advantage can directly benefit future mechanistic and therapeutic work that aims to either recapitulate or interfere with that functional change. Secondly, a fitness estimate provides information on the future growth trajectory of a clone, thus informing monitoring and risk stratification of patients, particularly, in the context of high risk clones. In blood, spliceosome mutations have high fitness estimates and do correlate with a high risk of AML (Fabre et al, Nature 2022). Thus, estimating at a population scale the fitness landscape of somatic mutations in tissues remains a valuable. Thirdly, discrepant estimates of selection by different methods may also yield important biological insights e.g. *DNMT3A* clone growth in adult life appears relatively slow, and yet dN/dS estimates suggest it to be consistently one of those under strongest selection. dN/dS gives an average estimate of selective advantage over the entire lifespan, whereas longitudinal studies of clone size estimate selection over the study period. Therefore, a possible explanation is transient strong selection during development and early life. Assessing selection by different methods and interpreting the results together can thus, be mechanistically informative.

At least half a dozen analyses of 100k+ participant UK Biobank cohorts have already been published. While this particular analysis is technically well done, the novelty of the findings is limited, and I don't think Nature Genetics is the appropriate venue for its publication.

We thank the Reviewer for their time and valuable comments. We hope we have addressed their concerns to improve the manuscript.

Reviewer #2:

Remarks to the Author:

The authors have done an excellent job of responding to the first round of reviews. I have no further comments.

We would like to thank the Reviewer for their time and valuable

comments. Reviewer #3:

Remarks to the Author:

I appreciate the author's thoughtful revision. I have no further comments.

We would like to thank the Reviewer for their time and valuable comments.

Final Decision Letter:

17th Apr 2024

Dear Dr. Nangalia,

I am delighted to say that your manuscript "Analysis of somatic mutations in whole blood from 200,618 individuals identifies pervasive positive selection and novel drivers of clonal hematopoiesis" has been accepted for publication in an upcoming issue of Nature Genetics.

Your paper will be published online after we receive your corrections and will appear in print in the next available issue. You can find out your date of online publication by contacting the Nature Press Office (press@nature.com) after sending your e-proof corrections.

Please note that *Nature Genetics* is a Transformative Journal (TJ). Authors may publish their research with us through the traditional subscription access route or make their paper immediately open access through payment of an article-processing charge (APC). Authors will not be required to make a final decision about access to their article until it has been accepted. Find out more about Transformative Journals

Authors may need to take specific actions to achieve compliance with funder and institutional open access mandates. If your research is supported by a funder that requires immediate open access (e.g. according to Plan S principles) then you should select the gold OA route, and we will direct you to the compliant route where possible. For authors selecting the subscription publication route, the journal's standard licensing terms will need to be accepted, including [a href="https://www.nature.com/nature-portfolio/editorial-policies/self-archiving-and-license-to-publish](https://www.nature.com/nature-portfolio/editorial-policies/self-archiving-and-license-to-publish). Those licensing terms will supersede any other terms that the author or any third party may assert apply to any version of the manuscript.

If you have not already done so, we invite you to upload the step-by-step protocols used in this manuscript to the Protocols Exchange, part of our on-line web resource, natureprotocols.com. If you complete the upload by the time you receive your manuscript proofs, we can insert links in your article that lead directly to the protocol details. Your protocol will be made freely available upon publication of your paper. By participating in natureprotocols.com, you are enabling researchers to more readily reproduce or adapt the methodology you use. [Natureprotocols.com](http://natureprotocols.com) is fully searchable, providing your

protocols and paper with increased utility and visibility. Please submit your protocol to <https://protocolexchange.researchsquare.com/>. After entering your nature.com username and password you will need to enter your manuscript number (NG-A63053R1). Further information can be found at <https://www.nature.com/nature-portfolio/editorial-policies/reporting-standards#protocols>

Sincerely,
Chiara

Chiara Anania, PhD
Associate Editor
Nature Genetics
<https://orcid.org/0000-0003-1549-4157>